# The Real Deal Behind the Artificial Appeal:
# Inferential Utility of Tabular Synthetic Data

**Alexander Decruyenaere** [*,1]  **Heidelinde Dehaene** [*,1]  **Paloma Rabaey**[2]  **Christiaan Polet**[1]
**Johan Decruyenaere**[1]  **Stijn Vansteelandt**[3]  **Thomas Demeester**[2]

[*]Joint first authors and corresponding authors
[1]Ghent University Hospital – SYNDARA research group, Belgium
[2]Ghent University – imec, Belgium
[3]Ghent University, Belgium

## Abstract

Recent advances in generative models facilitate the creation of synthetic data to be made available for research in privacy-sensitive contexts. However, the analysis of synthetic data raises a unique set of methodological challenges. In this work, we highlight the importance of inferential utility and provide empirical evidence against naive inference from synthetic data, whereby synthetic data are treated as if they were actually observed. Before publishing synthetic data, it is essential to develop statistical inference tools for such data. By means of a simulation study, we show that the rate of false-positive findings (type 1 error) will be unacceptably high, even when the estimates are unbiased. Despite the use of a previously proposed correction factor, this problem persists for deep generative models, in part due to slower convergence of estimators and resulting underestimation of the true standard error. We further demonstrate our findings through a case study.

## 1 INTRODUCTION

Data lie at the core of various disciplines that have a substantial impact on our daily life. Transparency and access to these data are therefore beneficial for an open society. However, alongside great opportunities, great precaution should be taken regarding the possible sensitive nature of these data and related privacy concerns (Nowok et al., 2016; Raghunathan, 2021).

Over the last decades, there is an increased awareness that conventional methods for anonymisation or deidentification are insufficient in terms of protecting the privacy and confidentiality of individuals (Bellovin et al., 2019; Ohm, 2009). Due to numerous well-documented statistical disclosure control failures resulting from the use of these methods, syn-

thetic data are being put forward as an alternative. The idea of creating synthetic data was first proposed by Rubin (1993) as an example of multiple imputation and has been further explored, culminating in extensive literature on this topic (Drechsler, 2011; Raab et al., 2016; Raghunathan, 2021; Raghunathan et al., 2003; Reiter, 2005). Although the idea itself is thus not novel, the advances in computing power and in the dynamic field of deep generative modelling caused a steep rise in research interest towards synthetic data (Drechsler & Haensch, 2023; Raghunathan, 2021; van Breugel et al., 2023).

Synthetic data are artificial data that (attempt to) mimic the original data in terms of statistical properties, without revealing individual records (Chen et al., 2021). As such, synthetic data might be able to replace the original data in analysis, while preserving the privacy of individual members of the original data and thereby fulfilling the regulatory privacy constraints (Kaloskampis et al., 2020; van Breugel et al., 2023; Z. Zhang et al., 2020). This could enable data sharing with the scientific community and therefore accelerate research, making synthetic data particularly appealing (van Breugel et al., 2023; Yan et al., 2022). Synthetic data can be generated using a broad spectrum of methods, ranging from statistical modelling techniques to highly innovative deep learning (DL) techniques such as Generative Adversarial Networks (GANs) and Variational Autoencoders (VAEs) (Endres et al., 2022; Hernandez et al., 2022; Nowok et al., 2016; Wan et al., 2017; Yan et al., 2022). These methods have also been extended to offer formal privacy guarantees by imposing differential privacy (Dwork & Roth, 2014) as an additional constraint during model training (Jordon et al., 2018; Xie et al., 2018; J. Zhang et al., 2017).

Consistent in the literature is the conclusion that the trade-off between taking steps to prevent disclosure of the identity of the individuals and preserving the data utility remains (Raghunathan, 2021). Moreover, there is a wide variation of metrics to assess data utility, often related to a quantification of how well the synthetic data resemble the real data or preserve their statistical information (e.g. in terms of distri-

*Accepted for the 40th Conference on Uncertainty in Artificial Intelligence* (UAI 2024).

bution, data types, or uni- and bivariate associations), also referred to as fidelity, and whether performance and feature selection in modelling tasks are congruent (El Emam, 2020; Ghosheh et al., 2022; Kaloskampis et al., 2020; Yan et al., 2022). Within the concept of data utility, we notice that inferential utility is often unmentioned, especially in the DL community (Drechsler & Haensch, 2023). Inferential utility captures whether a synthetic sample can be used to obtain valid estimates for a population parameter and to test hypotheses. Therefore, it describes whether one can make valid inferences concerning the population (Raghunathan, 2021). Wilde et al. (2021) argue that when synthetic data are used as if they were real data, inferential statements are only related to the synthetic and not the real data generating process, thereby compromising inferential utility. However, they only focus on differentially private synthetic data and state that the fundamental problem of inference is that the synthetic data generating process is misspecified by design, resulting from the additional constraints that are added to guarantee differential privacy. In our work, we will further prove empirically that the problem expands beyond differentially private synthetic data.

While Drechsler (2011), Raghunathan et al. (2003), Raghunathan (2021) and Räisä et al. (2023) developed procedures to obtain valid inferences from *multiple* synthetic datasets, we instead focus on inference from a *single* synthetic dataset, created by both statistical and DL techniques. This choice is motivated by previous research showing that the risk of disclosure increases with the number of synthetic datasets (Drechsler & Reiter, 2009; Klein & Sinha, 2015; Reiter & Mitra, 2009). The work of Raab et al. (2016) is closely related to the work presented here, since they derive expressions for the standard error (SE) of an estimator from a *single* synthetic dataset. However, they fail to take into account the implications of the regularisation bias prevalent in DL techniques (i.e. their bias-variance trade-off being optimised with respect to the prediction error instead of the error in the estimator).

Unfortunately, the regularisation bias introduced by data-adaptive DL techniques to prevent overfitting makes it impossible to guarantee close agreement between all functionals calculated on the real vs. synthetic data, thereby leaving overly optimistic impressions of data utility. Complex functionals involving higher-dimensional associations are arguably more vulnerable to this (Van der Laan & Rose, 2011). Moreover, naive SEs calculated on the synthetic data ignore the uncertainty (and regularisation bias) induced by the generative model. While this excess uncertainty and regularisation bias shrink with sample size (at different rates for different techniques (Brain & Webb, 1999; Hines et al., 2022)), they may be large relative to the size of naive SEs at each sample size, resulting in naive confidence intervals that (almost) never contain the population parameter. This excess variability is difficult to systematically account for

and, as far as we are aware, this has not yet been studied in the context of synthetic data generated by DL techniques.

In this work, we will focus on the *inferential utility* of *tabular* synthetic data. We identify three key contributions. First, we empirically investigate the behaviour of various estimators in terms of bias, SE and their convergence rates when estimated in synthetic data generated by both statistical and DL approaches. Second, we demonstrate how deviations from default behaviour in these properties lead to overly optimistic or even wrong conclusions through an inflation of the type 1 error rate. These issues are especially apparent for DL approaches. Finally, we show by means of a simulation and case study that the inferential utility of synthetic data remains compromised despite the use of a correction to the SE, as previously proposed by Raab et al. (2016). Overall, we aim to raise awareness that the current correction factors for the SE of an estimator are not routinely capable of capturing all added variability inherent to synthetic data generated by DL approaches.

The paper is organised as follows. Section 2 elaborates on the statistical properties that we examine in the context of inferential utility and the corrected SE proposed by Raab et al. (2016) for estimation with synthetic data. To assess the uncertainty in the estimates obtained from synthetic data and to explore their convergence rate, we conduct a simulation study. We also investigate the impact of deviations from the default behaviour of estimators on null hypothesis significance testing. Section 3 outlines our experimental setup and the generative models and statistical estimators considered. The findings of the simulation study are summarised and discussed in Section 4. Finally, to emphasise the relevance of this paper, we illustrate our key contributions by a case study using a well-known dataset in Section 5.

## 2 EVALUATING STATISTICAL PROPERTIES BASED ON SYNTHETIC DATA

In the literature, there is lack of consensus on the metrics that should be applied when evaluating synthetic data because of the complexity of synthetic data and the specific demands each use case has (Alaa et al., 2022; van Breugel et al., 2023; Yan et al., 2022). Most metrics are generally developed with the aim of assessing utility and/or privacy, where e.g. Yan et al. (2022) proposed a benchmarking framework that incorporates both facets. However, in this work, we investigate the impact of estimating population parameters from synthetic data, which may no longer have the same inferential utility when they are estimated as if the data were really observed. We deem it important to stress that our purpose is not to propose another utility measure, but rather to evaluate the inferential utility itself for different generative models. More specifically, focus lies on studying the

validity of estimators that are well-established on original data, but remain understudied in synthetic data, especially when these data are created by a DL approach.

The *estimands* considered in our simulation study range from the population mean to various regression coefficients. Commonly used estimators for these estimands (such as the sample mean and logistic regression coefficients) are further referred to as *estimators*, the obtained values of those estimators in a specific sample as *estimates*.

## 2.1 LARGE SAMPLE BEHAVIOUR OF ESTIMATORS

The purpose of the simulation study in Section 4 is to evaluate the quality of estimators of finite-dimensional parameters calculated on synthetic data. When an inferential statement is made, we rely on the test statistic and its properties to obtain a *p*-value. The formula for a test statistic is typically made up of the estimate divided by its SE. Therefore, we look at the empirical **bias** and **standard error (SE)** of the estimator by means of a simulation study, and how these evolve with increasing sample size. In standard statistical analyses, both are supposed to diminish as the sample size tends to infinity. It is typically seen that the **convergence rate** of the SE is of the order $1/\sqrt{n}$ while the bias converges faster (Lehmann & Casella, 2006). Such estimators are called roughly $\sqrt{n}$-consistent (with $n$ referring to the size of the original data). When this is not the case, standard statistical inference will be compromised. Therefore, we first assess how the estimators behave in terms of bias and SE when estimated in synthetic data. As indicated in Section 1, we foresee atypical behaviour for the estimators given the additional variability inherent to the generation process of the synthetic data and the regularisation bias, which should ideally be accounted for (Brain & Webb, 1999). Hence, we subsequently map the inferential repercussions of this deviant behaviour in the context of null hypothesis testing by quantifying the empirical **type 1 error rate**, i.e. the probability to find a significant effect when in truth there is none, and the empirical **power**, i.e. the probability to find a true significant effect.

## 2.2 MINIMAL CORRECTION FOR ESTIMATION

Even when synthetic data are generated based on a correct statistical model (i.e. without model misspecification) and without data-adaptive modelling (unlike DL methods), the regular expressions for the SE of estimators are insufficient when used in synthetic data. Within a statistical approach, the original sample is used to obtain a parametric representation of the dependency structure of the data. Based on these representations, synthetic samples are generated and then used to estimate the population parameters. When it is silently assumed that synthetic data can be treated as real data, as would be the case if standard expressions for the SE are used, this will lead to an underestimation of the SE due to ignoring the uncertainty in the generation process.

We foresee that the added variability of estimation based on synthetic data will diminish when the sample size of the original training sample increases. Intuitively, there is more uncertainty (and thus model variability) when a synthetic sample of e.g. 200 instances is created based on an original sample of 100 vs. $100\,000$ observations. Therefore, in the absence of both model misspecification and data-adaptive modelling, this added variability in a statistical approach will not induce large sample bias and will decrease with increasing sample size (Vansteelandt & Dukes, 2022). Consequently, when using a (pre-specified) parametric statistical approach to create synthetic data (where the size of the synthetic data is a fixed fraction $\in\,]0,1]$ of $n$) and an estimator that is $\sqrt{n}$-consistent on the original data, it is expected that the estimators will remain unbiased and $\sqrt{n}$-consistent (relative to the original data). It should be emphasised that this behaviour will not occur when synthetic data are created with a DL approach, due to the added variability and regularisation bias (with the latter not present in parametric statistical approaches) converging at slower rates.

In Appendix A, we provide an analytic derivation for a correction to the SE that is valid with a single synthetic dataset for any $\sqrt{n}$-consistent estimator $\hat\theta$. Originally proposed by Raab et al. (2016), the corrected SE is defined as follows:

$$\sigma_{\hat\theta,\,\text{corrected}} = \sigma_{\hat\theta,\,\text{naive}}\sqrt{1+\frac{m}{n}}, \qquad (1)$$

where $\sigma_{\hat\theta,\,\text{naive}}$ is the model-based SE of the estimator $\hat\theta$ in the synthetic data, $m$ the sample size of the synthetic data, and $n$ the sample size of the original data. This adaptation to the model-based SE will henceforth be referred to as the corrected SE. It is important to stress that the proof assumes a $\sqrt{n}$-consistent estimator when used in the original data. Therefore, Equation (1) is a *minimal* correction which does not account for the added variability resulting from the regularisation bias, making it insufficient when synthetic data are created using a DL approach. While Raab et al. (2016) implicitly assume $\sqrt{n}$-consistency and do not provide a formal derivation for the corrected SE with a single synthetic dataset, our contribution is to make this assumption explicit through a formal derivation, as well as to show empirically that it indeed does not sufficiently correct the SE in cases where $\sqrt{n}$-consistency cannot be guaranteed.

## 3 EXPERIMENTAL SETUP

To increase awareness that traditional statistical analyses, as well as corrected alternatives like the one discussed in the previous section, may fail when applied to synthetic data, we developed a general framework that will be used on toy data

(simulation study in Section 4) and real-world data (case study on Adult Census Income dataset in Section 5). In this section, we elaborate on this framework and the generators used to create synthetic data.

## 3.1 GENERAL FRAMEWORK

Some notation is introduced for the remainder of this paper, in line with the notation used in Stadler et al. (2022). A visualisation of the framework is given in Figure 1. We are interested in $\mathcal{R}$, a population of subjects where each data record $\boldsymbol{r} \in \mathcal{R}$ encompasses information on $p$ variables: $\boldsymbol{r} = (r_1, \ldots, r_p)$. These data follow an unknown joint probability distribution, $\mathcal{R} \sim \mathcal{D}_{\mathcal{R}}$. In reality, this population cannot be observed and hence a random sample is taken. We refer to this observed original data as $R \sim \mathcal{D}_{\mathcal{R}}^n$, which are $n$ independent data records from $\mathcal{D}_{\mathcal{R}}$, and these original data define the data distribution $\mathcal{D}_R$. This process is repeated $K$ times, further referred to as $K$ Monte Carlo runs.

Next, in the process of generating tabular synthetic data, a generative model will aim to learn a representation of the joint probability distribution $\mathcal{D}_R$ based on the original data $R$. Formally, a model training algorithm will learn a representation of the distribution $\mathcal{D}_R$, which is then denoted as $\mathcal{D}_{g(R)}$, and outputs a trained, yet stochastic generative model $g(R)$. Subsequently, synthetic data are generated based on this model $g(R)$. Synthetic data records are distributed according to $\mathcal{D}_{g(R)}$ and form the synthetic dataset $S = (\boldsymbol{s}_1, \ldots, \boldsymbol{s_m})$ of size $m$, $S \sim \mathcal{D}_{g(R)}^m$.

Finally, we examine the performance of diverse estimators when estimated in the $K$ synthetic datasets. This includes an evaluation of their bias and SE, their convergence rates, and some inferential metrics typically seen in null hypothesis testing (i.e. type 1 error rate and power).

## 3.2 SYNTHETIC DATA GENERATORS

We chose diverse data generation methods that are representative in terms of use and that enable us to examine the impact of added variability in the generation process and the extra layer of complexity due to regularisation bias. Following the categorisation suggested in Hernandez et al. (2022), we study both statistical (classical) approaches and DL approaches. In the following, a compact description of the applied methods is given, but a more detailed explanation of all these methods can be found in Appendix B.

The first implementation of a statistical approach, named **Synthpop**, relies on the Synthpop package for R, which provides a routine to generate synthetic data (Nowok et al., 2016). This framework encompasses both parametric and non-parametric methods to sequentially fit a series of conditional distributions, based on the observed data. We restrict ourselves to the default parametric method and pro-

vide information of the dependency structure of our data via specification of a directed acyclic graph (DAG). This defines the order of the sequence and which variables need to be included as predictors in the conditional models.

A second and third implementation of a statistical approach are based on Bayesian Networks (BNs) (Pearl, 2011). We opted to implement both a method where the dependency structure was pre-specified by the user through a DAG (**BN DAG**) and a method where the DAG was estimated automatically using the Chow-Liu algorithm (Chow & Liu, 1968) (**BN**). We refer to both methods as statistical approaches given that they rely on the Maximum Likelihood Estimator to estimate the conditional probability distributions. However, **BN** includes a data-adaptive component since the structure of the Bayesian Network is learned non-parametrically via the Chow-Liu algorithm.

Two commonly used generative DL approaches are GANs (Goodfellow et al., 2014) and VAEs (Kingma & Welling, 2013). We decided to focus on these methods, as they have been specifically adapted towards tabular data (Xu et al., 2019), and have been frequently used in recent literature on synthetic data (Akiya et al., 2024; Assefa et al., 2021; Bourou et al., 2021; Chalé & Bastian, 2022; El Emam et al., 2024; Figueira & Vaz, 2022; Liu et al., 2022; Rajabi & Garibay, 2022; Tao et al., 2021; van Breugel et al., 2023). Tabular data impose several challenges to the design of generative models, such as mixed data types, non-normality, and highly imbalanced categorical variables. To overcome these difficulties, Xu et al. (2019) propose **CTGAN** and **TVAE**. Training details for all DL approaches, including hyperparameter tuning, can be found in Appendix B.3. To study the effect of hyperparameter tuning on the inferential utility of synthetic data, we also considered untuned versions, further denoted as **Default CTGAN** and **Default TVAE**, where the hyperparameters were set to their default values as suggested by the Synthcity library (Qian et al., 2023).

Both privacy and utility are essential for synthetic data and their trade-off should be optimised. In our study, inferential utility was the starting point and as such, we did not formally define and assess privacy. However, by imposing differential privacy as an additional constraint during model training, some generative models provide formal privacy protection guarantees. Complementary to the approaches described above, we additionally study three state-of-the-art privacy-focused generators, i.e. **PrivBayes** (J. Zhang et al., 2017), **DP-GAN** (Xie et al., 2018), and **PATE-GAN** (Jordon et al., 2018). All results pertaining to this class of generative models are presented in Appendix E, confirming that the conclusions of our study extend to privacy-focused generators.

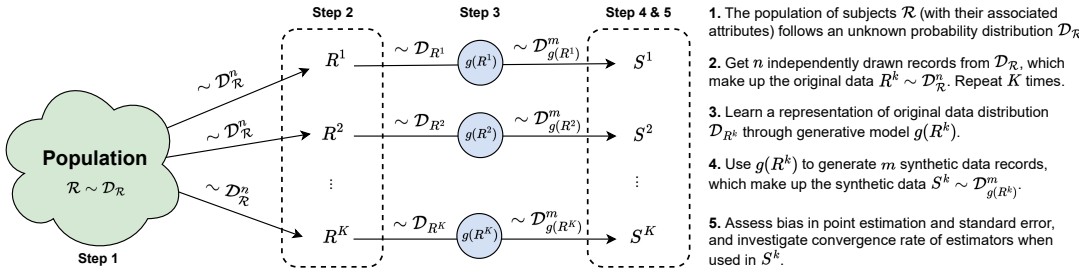

Figure 1: General experimental framework, applied in both the simulation study and case study.

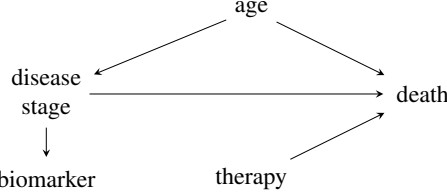

Figure 2: DAG for the variables in the simulation study.

# 4 SIMULATION STUDY

In order to assess the finite sample performance of the different estimators and the correction for the SE proposed by Raab et al. (2016) in the context of synthetic data, a Monte Carlo simulation study is performed. In line with the experimental setup from Section 3.1, the sections below introduce the data on population level (Section 4.1), and list the experimental details of our simulation study, after which we dive into the results (Section 4.2).

## 4.1 DATA GENERATING MECHANISM

We opted to work with low-dimensional tabular data given their frequent use in applied medical research. Commonly used regression models were taken into account when choosing the nature of the variables. We included a mix of continuous (normally distributed or skewed), binary, and ordinal variables. To obtain these requirements and reflect a generic clinical setting, the data generating process consists of the following five variables: *age* (continuous with a normal distribution), *disease stage* (ordinal with four categories), *biomarker* (continuous with a skewed distribution), *therapy* (binary), and *death* (binary). A DAG representing the dependency structure is shown in Figure 2. We refer to Appendix C.1 for the exact data generating mechanism.

## 4.2 SPECIFICATIONS AND RESULTS

In line with Figure 1, we simulate $n$ independent records from the population $\mathcal{R}$ that form the observed original data $R$. This process is repeated $K = 200$ times, with the sample size $n$ varying log-uniformly between 50 and 5000 (i.e. $n \in \{50, 160, 500, 1600, 5000\}$). Per generation method as introduced in Section 3.2, a generative model $g(R^k)$ is trained, from which $m$ synthetic data records are sampled. In our study, we set $m = n$ to retain the dimensionality of the original data and to facilitate an equal comparison between original and synthetic data. This process results in 200 synthetic datasets $S$ for each of the generator methods and each value of $n$.

We then evaluate a variety of statistical estimators in these synthetic datasets. Motivated by commonly used analyses in applied medical research, and the variety of mixed data types in our setup, we opted to work with the following estimators: mean, proportion, and regression coefficients from a main effects proportional odds cumulative logit model (effect of *age* on *disease stage*), a main effects gamma regression model (effect of *disease stage* on *biomarker*), and a main effects binomial logistic regression model (effect of *age*, *disease stage* and *therapy* on *death*).

We now present the results of our simulation study. In Section 4.2.1 we evaluate the quality of our synthetic data. Then, Section 4.2.2 investigates the bias and SE of the estimators, after which Section 4.2.3 discusses their convergence rate. Finally, Section 4.2.4 addresses the impact of the atypical behaviour of the estimators on null hypothesis testing. The code to reproduce all results is available on Github: `https://github.com/syndara-lab/inferential-utility`.

### 4.2.1 Quality of Synthetic Data

The average inverse of the Kullback–Leibler divergence (IKLD) is often used to assess the statistical similarity between distributions (Espinosa & Figueira, 2023). This metric was used to tune the hyperparameters of **CTGAN** and **TVAE**. The average IKLD over all Monte Carlo runs is presented per generator in Table C1 in the appendix, where the tuned **CTGAN** and **TVAE** have higher IKLD than their default versions. However, the statistical approaches still seem to perform slightly better. Additional analyses on synthetic data quality are included in Appendix C.2.

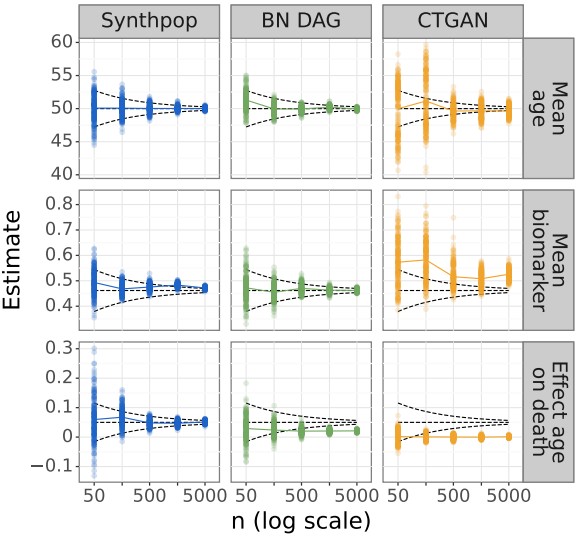

Figure 3: The horizontal dashed line represents the population parameter and each dot is an estimate per Monte Carlo run (200 dots in total per value of $n$). The dashed funnel indicates the behaviour of an unbiased and $\sqrt{n}$-consistent estimator based on observed data.

### 4.2.2 Bias and Standard Error

Figure 3 depicts the estimates for a selection of estimators (the sample mean of $age$ and $biomarker$, and the logistic regression coefficient of $age$ on $death$) and generators (**Synthpop**, **BN DAG** and **CTGAN**). Figures C1 and C2 in the appendix show these for all estimators and generators. Each dot is an estimate per Monte Carlo run and the true population parameters are represented by the horizontal dashed line. This figure allows a qualitative assessment of two key properties of estimators: **empirical bias** (i.e. the average difference between the estimates and the population parameter, as represented by the solid line) and **empirical SE** (i.e. the standard deviation of the estimates, as indicated by the vertical spread of the estimates). Table 1 lists the same information numerically, summarising the relative bias ($\text{RE}_{\hat{\theta}}$) and the relative underestimation of the empirical SE by the naive model-based SE ($\text{RE}_{\hat{\sigma}_{\hat{\theta}}}$). Tables C4 and C5 in the appendix show these metrics for all estimators and generators.

Ideally, both the bias and SE converge to zero as the sample size grows larger. The convergence rate conveys the rate at which this happens. The funnel in Figure 3 represents the default behaviour of an unbiased estimator based on original data of which the SE diminishes at a rate of $1/\sqrt{n}$. We observe that the bias and SE of estimators based on synthetic data often deviate from this default behaviour, the extent of which differs between generative models.

First, generative model misspecification will introduce bias.

This is for example the case with **Synthpop**, where the sample mean of $biomarker$ consistently overestimates the population mean. $Biomarker$ is gamma-distributed in the original data, but **Synthpop** fails to reconstruct this marginal distribution since it uses (by default) an ordinary least squares regression model during the generation process, leading to reconstruction error. Generative model misspecification also occurs for the logistic regression coefficients based on synthetic data generated by **BN DAG**, resulting in bias towards the null effect. This arises from the full discretisation of the continuous variable $age$, along with discrete variables $stage$ and $therapy$, during the conditional generation of $death$, introducing non-negligible overfitting bias in the latter. By contrast, the parsimonious nature of the Chow-Liu algorithm applied in **BN** provides some immunity against the impact of this discretisation on overfitting. **CTGAN**, despite being more flexible yet tuned to prevent overfitting, also fails to adequately fit the joint distribution in our simulation study, leading to bias for several estimators including a biased null effect of $age$ on $death$. We also observe some bias for the other DL approaches in Table C5 in the appendix. Note that this might be partially attributed to the fact that these methods do not receive prior knowledge on the dependencies between the variables, while **Synthpop** and **BN DAG** do. Further, this model misspecification is not captured well by the IKLD metric described in Section 4.2.1. Moreover, the model misspecification for **CTGAN** could even directly result from the tuning objective being based on the average IKLD, because this will not prioritise the preservation of multivariate relations as the divergences are only calculated per single variable and then averaged across all variables.

Second, the empirical SEs are larger for synthetic data than for original data and may vary over generative models. Larger SEs reflect the additional (predictive) uncertainty in the generation of synthetic data, which seems most pronounced with DL approaches. This uncertainty is discarded in the naive use of model-based SEs, leading to underestimation of the empirical SE, as is evident from Table 1.

### 4.2.3 Convergence Rate

Assuming a power law $n^{-a}$ in convergence rate for the empirical bias and SE, we estimated the exponent $a$ from five logarithmically spaced sample sizes $n$ between 50 and 5000, shown for a selection of estimators in Table 2. The results (with 95% confidence interval) for all estimators and generators can be found in Tables C6 and C7 and Figure C3 in the appendix. Standard statistical analysis assumes that the bias converges faster than the SE with the latter diminishing at a rate of $1/\sqrt{n}$. The corrected SE proposed by Raab et al. (2016), though taking into account the added variability of the synthetic data generating process, still relies on the same assumption, thus remaining invalid for deviating convergence rates.

Table 1: Relative error (RE) for **Synthpop**, **BN DAG** and **CTGAN** for a selection of estimators, averaged over 200 Monte Carlo runs. $RE_{\hat{\theta}}$ is the relative bias of the estimates $\hat{\theta}$. $RE_{\hat{\sigma}_{\hat{\theta}}}$ is the RE between the naive model-based ($\hat{\sigma}_{\hat{\theta},naive}$) and the empirical standard error. Positive and negative values indicate a relative over- and underestimation.

| | Synthpop | | | | BN DAG | | | | CTGAN | | | |
| | $RE_{\hat{\theta}}$ (%) | | $RE_{\hat{\sigma}_{\hat{\theta}}}$ (%) | | $RE_{\hat{\theta}}$ (%) | | $RE_{\hat{\sigma}_{\hat{\theta}}}$ (%) | | $RE_{\hat{\theta}}$ (%) | | $RE_{\hat{\sigma}_{\hat{\theta}}}$ (%) | |
| Estimator | $n=50$ | $n=5000$ | $n=50$ | $n=5000$ | $n=50$ | $n=5000$ | $n=50$ | $n=5000$ | $n=50$ | $n=5000$ | $n=50$ | $n=5000$ |
|---|---|---|---|---|---|---|---|---|---|---|---|---|
| Mean age | 0.15 | 0.03 | -40.31 | -26.55 | 2.59 | -0.10 | -6.17 | -2.32 | -0.03 | -0.73 | -46.38 | -78.85 |
| Mean biomarker | 6.86 | 2.16 | -14.23 | 1.16 | 2.32 | 0.05 | -21.31 | -26.83 | 24.14 | 13.87 | -35.77 | -76.00 |
| Proportion therapy | 0.18 | -0.12 | -33.26 | -31.47 | -25.28 | -0.93 | 7.00 | -4.56 | -0.42 | -0.18 | 131.41 | -56.79 |
| Proportion death | 31.39 | -2.35 | -25.40 | -14.07 | 7.27 | 3.23 | 45.38 | 25.23 | 6.04 | 5.29 | -9.52 | -47.44 |
| Effect age on death | 19.07 | 2.35 | -33.90 | -30.04 | -41.76 | -57.81 | -9.98 | -3.13 | -96.42 | -97.81 | -9.41 | -1.18 |
| Effect therapy on death | 38.53 | -2.74 | -32.78 | -30.45 | -50.16 | -55.24 | -11.63 | -15.83 | -104.72 | -101.02 | 17.50 | -13.89 |

Table 2: Estimated exponent $a$ for the power law convergence rate $n^{-a}$ for empirical bias and standard error (SE).

| | Generator | | | |
| Estimator, bias/SE | Original | Synthpop | BN DAG | CTGAN |
|---|---|---|---|---|
| Mean age | 0.64 / 0.49 | 0.38 / 0.53 | 0.45 / 0.49 | -0.42 / 0.40 |
| Mean biomarker | 0.47 / 0.48 | 0.10 / 0.51 | 0.81 / 0.49 | 0.18 / 0.34 |
| Proportion therapy | 0.42 / 0.50 | 0.12 / 0.50 | 0.81 / 0.46 | 0.10 / 0.19 |
| Proportion death | 1.24 / 0.51 | 0.45 / 0.52 | 0.02 / 0.48 | 0.04 / 0.41 |
| Effect age on death | 0.76 / 0.56 | 0.52 / 0.59 | -0.07 / 0.57 | -0.00 / 0.47 |
| Effect therapy on death | 0.70 / 0.53 | 0.52 / 0.56 | -0.02 / 0.53 | 0.00 / 0.51 |

As shown in the table, the empirical SE of estimators based on original data indeed converges at a rate of $1/\sqrt{n}$ (i.e. $a_{SE} \approx 0.5$). Fully parametric generative models are also expected to yield estimators of which the SE decreases at a rate of $1/\sqrt{n}$. This seems confirmed by **Synthpop**, **BN DAG**, and **BN**. By contrast, the SEs produced by the more data-adaptive DL approaches converge much slower (i.e. $a_{SE} << 0.5$). The slower-than-$\sqrt{n}$-convergence leads to a progressively increasing underestimation of the empirical SE by the model-based SE (which assumes $\sqrt{n}$-convergence) as the sample size grows larger, as seen in Table 1. Furthermore, as opposed to default behaviour, the bias converges slower than the SE ($a_{bias} \leq a_{SE}$) for some estimators and generators. This is problematic, as elaborated on in the next section.

#### 4.2.4 Null Hypothesis Significance Testing

The null hypothesis significance testing (NHST) framework typically uses an estimate divided by its associated (un)certainty (reflected by the SE) as test statistic to reject a null hypothesis. We foresee the following problems with NHST on synthetic data. First, if the bias of the estimator converges slower than its SE, the test statistic will become more extreme, thereby increasing the type 1 error rate. Second, even if the bias converges faster than the SE, the naive

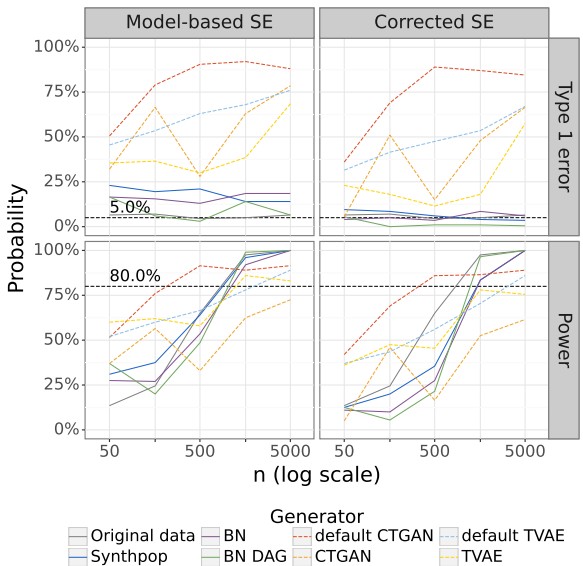

Figure 4: Type 1 error rate and power of a one-sample t-test at $\alpha = 5\%$ for the population mean of $age$ with naive model-based and corrected standard errors (SEs).

model-based SEs are still optimistically small, also inflating the type 1 error rate. Both scenarios are highly concerning, since they will lead to a flurry of false-positive findings. Acknowledgement of the extra uncertainty by using a larger yet valid SE will control the type 1 error rate at the nominal level. In turn, this will decrease the power, reflecting the loss of information when working with synthetic data.

Figure 4 shows the empirical type 1 error rate and power of a one-sample t-test at $\alpha = 5\%$ for the population mean of $age$, separately tested with naive model-based SEs and corrected SEs as suggested by Raab et al. (2016). For the type 1 error rate, the null hypothesis states that the population mean of $age$ is equal to the ground truth, whereas for the power, it states that the population mean of $age$ is equal to $98\%$ of the ground truth.

Hypothesis tests with naive SEs lead to type 1 error rates larger than $5\%$: the more the empirical SE is underestimated by the naive model-based SE (as is especially the case for the DL approaches with increasing $n$), the larger the inflation of the type 1 error rate. Use of corrected SEs will control the type 1 error rate at approximately $5\%$, but only for statistical approaches. However, this comes with a loss of power, a trade-off most pronounced in small sample sizes. More importantly, the corrected SE does not sufficiently account for the predictive uncertainty of the DL approaches, so the type 1 error rate remains uncontrolled for. It is essential to highlight that the low inferential utility of synthetic data generated by DL approaches will be observed regardless of whether the estimator is biased, so this limitation cannot be explained in terms of bias or poor performance of one DL approach.

## 5 CASE STUDY

To illustrate our findings and their implications for the applied researcher, we perform a case study on the Adult Census Income dataset (Becker & Kohavi, 1996) following the framework discussed in Section 3. The dataset with $45\,222$ complete cases constitutes our *population*. We assume that the researcher only has access to a limited *sample* of 5000 observations. In order to share their data without privacy issues, the researcher generates a synthetic dataset, with $n = m = 5000$, once using the statistical method **Synthpop**, and once by training a **Default CTGAN**. For this case study, it was computationally too intensive to create synthetic values for variables with a large number of categories based on a parametric method in **Synthpop**. Therefore, we used the non-parametric method *CART* for these unordered categorical variables. Additional analyses on synthetic data quality are included in Section D.1 of the appendix.

For simplicity, we assume the researcher's interest lies in inferring the population mean of $age$, and the effect of $age$ on $income$ (estimated via a logistic regression model) from a single synthetic dataset. When an estimate for these targets is obtained, an inferential statement can be made by using a $95\%$ confidence interval (CI). A CI for the population parameter is constructed by using the estimate and its model-based SE obtained from that synthetic set of 5000 instances, in our case resulting in 200 CIs. If we repeated the construction of CIs infinitely, then $95\%$ of the $95\%$ CIs should by definition cover the population parameter. Figure 5 depicts the first 15 CIs obtained from both original and synthetic samples for the effect of $age$ on $income$. The vertical dashed lines represent the true parameter value as obtained from the *population*. Comparing the point estimates with this dashed line, one can see that the estimates obtained in the synthetic samples are positioned around the true population parameter, but that the variability is much higher than in the original samples. This is in accordance with the results

Table 3: Results for the case study: relative error (RE) for the estimates $\hat{\theta}$ and the model-based standard errors (SEs) $\hat{\sigma}_{\hat{\theta}}$, and empirical coverage of $95\%$ confidence intervals with the model-based (Cov) and corrected SE (Cov$_{\text{corr}}$) (in %).

| | Original | | Synthpop | | Default CTGAN | |
|---|---|---|---|---|---|---|
| **Estimator** | RE$_{\hat{\theta}}$ | RE$_{\hat{\sigma}_{\hat{\theta}}}$ | RE$_{\hat{\theta}}$ | RE$_{\hat{\sigma}_{\hat{\theta}}}$ | RE$_{\hat{\theta}}$ | RE$_{\hat{\sigma}_{\hat{\theta}}}$ |
| Mean age | 0.09 | 4.76 | -0.03 | -28.98 | -1.58 | -93.65 |
| Effect age on income | -0.09 | 20.39 | -1.96 | -50.11 | 3.87 | -56.51 |
| **Estimator** | Cov | Cov$_{\text{corr}}$ | Cov | Cov$_{\text{corr}}$ | Cov | Cov$_{\text{corr}}$ |
| Mean age | 95.50 | – | 83.50 | 95.00 | 12.18 | 13.20 |
| Effect age on income | 98.00 | – | 70.50 | 87.00 | 59.39 | 77.16 |

from the simulation study and hence endorses the claim that the SE should incorporate the extra variability caused by the synthetic data generation process. Table 3 also shows that the SE for all estimators is now highly underestimated when estimated in the synthetic samples.

More strikingly, we find that for synthetic data created with **Default CTGAN**, only 8 out of 15 CIs depicted in Figure 5 contain the true parameter value. This is also quantified by the low empirical coverage levels reported in Table 3, and is even more pronounced for the mean of $age$ (see Figure D1 in the appendix). Combining all results from Table 3 (i.e. limited bias in point estimations but substantial underestimation of the SE and thus low empirical coverage levels), we can state that naive CIs based on a synthetic sample are too narrow (or *permissive*), hence overestimating the confidence about the estimated mean of $age$ and effect of $age$ on $income$ in the *population*. Using the corrected SE improves the coverage to some extent, as seen in Figure 5 and Table 3, but this is still highly insufficient for synthetic data created with **Default CTGAN**. For **Synthpop**, we observe that the correction works properly on the estimator for the population mean of $age$, but fails to fully elevate the empirical coverage to the nominal level for the effect of $age$ on $income$. These mixed results were expected, given that $age$ was a root node and therefore synthesised based on bootstrap samples. Conversely, $income$ was synthesised based on all other previously (non-parametrically) synthesised variables. This reinforces our claim that the corrected SE should be seen as a minimal correction, depending on the used synthetic data generator.

## 6 DISCUSSION

We conducted a simulation study to quantify how the naive use of statistical estimators in synthetic data (which silently assumes that these data can be treated as if directly observed) compromises the inferential utility. We studied both statistical approaches and deep learning techniques. We empirically confirmed that increased variability in the estimates

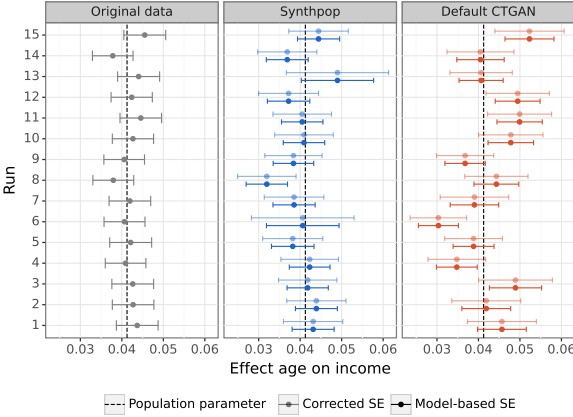

Figure 5: Empirical coverage of $95\%$ confidence intervals for effect of $age$ on $income$, with model-based and corrected standard error (SE).

leads to an underestimation of their standard error, supporting previous claims by Raab et al. (2016). Our simulation study moreover revealed the slower-than-expected convergence rate of both bias and standard error, which was most pronounced for deep learning approaches.

We also tested the corrected standard error as proposed by Raab et al. (2016) for applications with only one synthetic dataset, and demonstrated that this adaptation does not sufficiently capture all added variability in the case of deep learning approaches. We argued that this is due to an extra layer of complexity introduced by their regularisation bias, which cannot readily be expressed analytically and therefore remains unaccounted for. Furthermore, the corrected standard error relies on the classical assumptions concerning the root-$n$ convergence rate of estimators, which were shown to be unfulfilled for the deep learning approaches studied in this work. The fact that these approaches fail to offer the specific guarantee of root-$n$ convergence implies that the corrected standard error will never approximate the empirical standard error, even when increasing the synthetic sample size $m \rightarrow +\infty$. Therefore, at present, this renders them not useful for statistical inference, despite their flexibility allowing them to better approximate a more complex joint distribution of the original data.

The impact of these deviations from default behaviour becomes apparent when evaluating metrics in the context of null hypothesis testing, often the prime interest of applied researchers. A naive use of synthetic data leads to an inflation of false-positive findings, which can be controlled for to some extent with the use of the corrected standard error, though only for data generated using parametric statistical approaches. This comes at a cost in terms of power loss, which is an inevitable trade-off. These practical implications were further cemented in our case study.

The broader implications of this work for an applied re-

searcher come into play at every stage of employing synthetic data. First of all, a reader who consults (published) work that is based on synthetic data should interpret naive analysis with caution. As empirically proven in this paper, standard confidence intervals and p-values obtained on synthetic data may drastically underestimate the uncertainty in synthetic data. Second, the analyst who only has access to the synthetic data should use an adequate correction for the standard error of an estimator when making inferential statements. However, we have shown that the current correction factors are not capable of capturing all added variability inherent to synthetic data generated by deep learning approaches. We deem it difficult to obtain a generic correction for deep learning approaches, since the uncertainty associated with their regularisation bias cannot readily be expressed analytically. Therefore, we can conclude that the original data holder who creates synthetic data, must know what analysis will be done on these data. If the goal is inference, we advise to use a parametric generation method, since the corrected standard errors are proven to be sufficient only in these settings.

Limitations of our study include the low-dimensional setting, for which deep learning approaches might be less suited. Still, the fact that deviant behaviour is already observed for a wide range of statistical estimators given the use of such a simple data generating mechanism, raises questions about what can be expected in larger-scale applications. We especially notice that deep learning approaches fall short. Our study did not cover more recently developed deep generative models, such as diffusion-based models or even large language models. While these models are popular, they are less well-established in the domain of tabular synthetic data than **CTGAN** and **TVAE**. Still, we expect the same problems to occur, since all deep learning approaches are designed to optimally balance bias and variance only w.r.t. a chosen criterion (like prediction error). Therefore, none of them can guarantee that an optimal trade-off is made simultaneously w.r.t. the mean squared error for all possible estimators.

To improve the inferential utility of synthetic data created by deep learning approaches, we propose the following ideas for future research. First, building on insights from the literature on debiased and targeted machine learning (Chernozhukov et al., 2018; van der Laan & Rose, 2011), there may be potential to eliminate bias by targeting synthetic data generators towards the considered statistical analysis. Second, importance weighting methods could be developed, similarly to what Ghalebikesabi et al. (2022) proposed in the context of noise-related bias in differentially private synthetic datasets.

**Acknowledgements**

This research was funded by a grant received from the Fund for Innovation and Clinical Research of Ghent University Hospital. Paloma Rabaey's research is funded by the Research Foundation Flanders (FWO-Vlaanderen) with grant number 1170122N. This research also received funding from the Flemish government under the "Onderzoeksprogramma Artificiële Intelligentie (AI) Vlaanderen" programme.

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

# Appendix

**Alexander Decruyenaere** [*,1]  **Heidelinde Dehaene** [*,1]  **Paloma Rabaey**[2]  **Christiaan Polet**[1]

**Johan Decruyenaere**[1]  **Stijn Vansteelandt**[3]  **Thomas Demeester**[2]

[*]Joint first authors and corresponding authors
[1]Ghent University Hospital – SYNDARA research group, Belgium
[2]Ghent University – imec, Belgium
[3]Ghent University, Belgium

## A   DERIVATION OF CORRECTED STANDARD ERROR

In this proof, we study the large sample behaviour of a $\sqrt{n}$-consistent estimator for a scalar parameter $\theta$ calculated on the synthetic data. Let $P$ refer to the true distribution of the data. Then $\theta$ can be viewed as a functional of $P$; we will denote it $\theta(P)$. For instance, the population mean of an outcome $Y$ can be written as $\theta(P) = \int y dP(y)$. The observed data forms a sample from the distribution $P$. They enable us to obtain an estimator of $P$, denoted $P_n$, so that estimators of $\theta$ can be viewed as the functional $\theta(.)$ evaluated at $P_n$: $\theta(P_n)$ (Bickel et al., 1993). For instance, with $P_n$ the empirical distribution of the data, which assigns point mass to each data point, we have that $\theta(P_n) = \int y dP_n(y) = n^{-1} \sum_{i=1}^{n} Y_i$. The fact that we consider $\sqrt{n}$-consistent estimators implies that $E\{\theta(P_n)\} = \theta(P) + o(n^{-1/2})$ and that $\text{Var}\{\theta(P_n)\} = \sigma^2(P)/n + o(n^{-1})$ for some constant $\sigma^2(P)$ (which need not represent the outcome variance).

When using synthetic data, we first construct an estimator $\hat{P}$ of $P$, next sample $m$ independent data from $\hat{P}$ and finally obtain an estimator of $\hat{P}$, denoted $\hat{P}_m$, so that the estimator obtained from the synthetic data can be written as $\theta(\hat{P}_m)$. We then have that

$$
\begin{aligned}
\text{Var}\left\{\theta(\hat{P}_m)\right\} &= E\left[\text{Var}\left\{\theta(\hat{P}_m)|\hat{P}\right\}\right] + \text{Var}\left[E\left\{\theta(\hat{P}_m)|\hat{P}\right\}\right] \\
&= E\left[\sigma^2(\hat{P})/m + o(m^{-1})\right] + \text{Var}\left[\theta(\hat{P}) + o(m^{-1/2})\right] \\
&= \sigma^2(P)/m + o(n^{-1}m^{-1}) + o(m^{-1}) + \sigma^2(P)/n + o(m^{-1}) + o(n^{-1}) \\
&= \sigma^2(P)\left(\frac{1}{m} + \frac{1}{n}\right) + o(m^{-1}) + o(n^{-1}).
\end{aligned}
$$

Here, we use that $\theta(.)$ and $\sigma^2(.)$ are smooth functionals of $P$ (in the sense of being path-wise differentiable parameters (Hines et al., 2022)) and $\hat{P}$ being a $\sqrt{n}$-consistent estimator of $P$ (which will generally be satisfied when a parametric synthetic data generation method is used, but not otherwise). We conclude that the standard error of the estimator for $\theta$ as obtained on synthetic data can be approximated in large samples as

$$
\sigma(P)\sqrt{\frac{1}{m} + \frac{1}{n}};
$$

note that 'large sample' here refers to both the original and synthetic data size being 'large'. The fact that a naive analysis will deliver a standard error equal to $\sigma(P)/\sqrt{m}$, explains the correction reported in the main text.

*Accepted for the 40$^{th}$ Conference on Uncertainty in Artificial Intelligence* (UAI 2024).

# B  SYNTHETIC DATA GENERATION METHODS

We elaborate on the generative models used to create synthetic data in our study. These are split into statistical approaches and deep learning (DL) approaches, following the categorisation suggested in Hernandez et al. (2022). All models were trained on our internal cluster using a single GPU (NVIDIA Ampere A100; only utilised by our DL methods) and eight CPUs (AMD EPYC 7413), taking less than 24 hours to complete.

## B.1  STATISTICAL APPROACHES

Our first statistical approach **Synthpop** uses the synthetic data generation framework built into the R package Synthpop (Nowok et al., 2016). Specifically, we rely on its default parametric method. In this approach, the user can provide the assumed dependencies between the variables in the form of a Directed Acyclic Graph (DAG), whose topological ordering prescribes the sequential order in which to synthesise the variables. In our simulation study, the true structure of the DAG as depicted in Figure 2 is provided. In our case study, we work with the Adult Census Income Dataset, for which the true dependency structure between the variables is unknown. In this case, we do not pass a DAG to the Synthpop method, which then falls back on using the arbitrary ordering of the columns in the dataset ($age$ first, $income$ last) as a variable ordering.

Once the sequential ordering is fixed, each variable is modelled by fitting a parametric or non-parametric representation based on the original data $R$, conditionally on its parent variables (except for the root nodes, which come first in the sequence and do not have any parents, and are generated using bootstrap samples). The type of model used to fit each representation is based on the data type of the considered variable. By default, Synthpop uses distribution-preserving linear regression, logistic regression, unordered polytomous regression, and ordered polytomous regression models for continuous, binary, unordered categorical, and ordered categorical variables, respectively. These synthesising methods were used in our simulation study, whereas in the case study classification and regression trees (non-parametric) were used for the generation of unordered categorical variables. This is because the Adult Census Income dataset contains multiple variables with many categories that cannot be readily fitted by the default unordered polytomous regression model.

Parallel to the fitting of each conditional distribution, synthetic data are generated for that particular variable. The exact implementation of this procedure depends on the assumed model and is based on the synthesised values of all variables preceding it in the sequential ordering. In addition, the process of generating data can be based on either a 'proper' or 'simple' synthesis, referring to whether each method samples from the posterior distribution of the parameters of the conditional models or not, respectively. In both the simulation study and case study, we opted for simple synthesis. For a more detailed explanation of Synthpop, we refer the interested reader to the package's source code and Nowok et al. (2016).

Both the second and third statistical approach are based on creating synthetic data through Bayesian Networks (BNs). We implement both a method where the DAG is pre-specified by the user (**BN DAG**) and a method where the DAG is estimated by using an algorithm (**BN**). In the former case, we again provide the true DAG structure as depicted in Figure 2 for our simulation study. In the latter case, the unknown DAG is estimated based on a tree search using the Chow-Liu algorithm (Chow & Liu, 1968) (note that this Bayesian approach is therefore not a 'pure' statistical method). For both Bayesian Network implementations, the conditional probability distributions (CPDs) are estimated using Maximum Likelihood Estimation (MLE) and synthetic data are generated via forward sampling. More information on the algorithm and estimators can be found in the documentation of PGMPY (Ankan & Panda, 2015).

Both **BN** and **BN DAG** are included in order to investigate whether the availability of the correct DAG would result in better performance (e.g. less variability in estimators) compared with a BN that does not have prior knowledge and needs to rely on data-adaptive DAG discovery. In many practical settings, the (full) DAG cannot be provided upfront since causal relationships between the variables are unknown. In those cases, dependency structure discovery methods like the Chow-Liu algorithm are often used to recover the DAG. Note that the performance of a model that follows this paradigm is upper bounded by the performance of the model that receives the DAG upfront.

There are multiple differences between **Synthpop** and the **BN** (with or without DAG) implementation. In a BN, a joint probability is obtained through factorisation. When all variables are discrete, natural estimates for the CPDs are the relative frequencies, which coincide with the MLE of a multinomial model. Within our BN implementation, continuous variables undergo discretisation. It is possible to avoid this, but practically this means imposing a linear Gaussian CPD for all variables, including the discrete ones, which undermines the representation power of the BN. In Synthpop, the joint distribution is also defined in terms of a series of conditional distributions. With its parametric methods, Synthpop imposes a specific

distribution and parametric regression model depending on the variable type. Therefore, the likelihood can now be written as a function of these regression models, instead of just the multinomial likelihood function seen in BN, and the corresponding parameters of these regression models are estimated via MLE. Depending on the variable type, different parametric models are possible, as opposed to BNs, where the distribution is either multinomial for discrete variables or Gaussian when (non-discretised) continuous variables are included in the mix. Thus, the difference between **Synthpop** and **BN** (with or without DAG) lies in the flexibility of the assumed parametric distribution and the way each method deals with mixed variable types.

## B.2 DEEP LEARNING APPROACHES

We focus on two commonly used deep generative models, namely Generative Adversarial Networks (GANs) (Goodfellow et al., 2014) and Variational Autoencoders (VAEs) (Kingma & Welling, 2013).

A GAN consists of two competing neural networks, a generator and discriminator, and aims to achieve an equilibrium between both (Hernandez et al., 2022). This translates to a mini-max game, since the generator aims to minimise the difference between the real and generated data, while the discriminator aims to maximise the possibility to distinguish the real and generated data (Goodfellow et al., 2014). We use the **CTGAN** implementation that was designed specifically for tabular data, proposed by Xu et al. (2019).

A VAE is a deep latent variable model, consisting of an encoder and a decoder (Kingma & Welling, 2013). The encoder models the approximate posterior distribution of the latent variables given an input instance, whereby typically a standard normal prior is assumed for the latent variables. The decoder allows reconstructing an input instance, based on a sample from the predicted latent space distribution. Encoder and decoder can be jointly trained by maximising the *Evidence Lower BOund* (ELBO), i.e. the marginal likelihood of the training instances. Maximising the ELBO corresponds to minimising the KL-divergence between the predicted latent variable distribution for a given input instance and the standard normal priors, and minimising the reconstruction error of the input instance at the decoder output. Once again, we use the tabular implementation of a VAE (**TVAE**) proposed by Xu et al. (2019).

As these DL approaches are very expressive models, especially compared to the low-dimensional data that were used in the simulation study, tuning these models is an important step towards prevention of overfitting. For this reason, the next section outlines the strategy followed to tune the hyperparameters of our **CTGAN** and **TVAE** models. To study the effect of hyperparameter tuning, we also considered untuned versions of both DL methods (**Default CTGAN** and **Default TVAE**). Here, we used the implementation from the Synthcity library, with the hyperparameters set to their default values (Qian et al., 2023).

## B.3 HYPERPARAMETER TUNING

Since Synthcity's implementation did not allow us to tune all hyperparameters we wanted, we implemented an extended version of the CTGAN and TVAE modules, where we used Synthetic Data Vault's implementation as a baseline (Patki et al., 2016). This made it possible to tune additional regularisation hyperparameters: generator and discriminator dropout were added to the CTGAN module, and encoder and decoder dropout were added to the TVAE module. Note that SDV implements the CTGAN and TVAE modules as originally proposed by Xu et al. (2019), where a cluster-based normaliser is used to preprocess numerical features.

**Hyperparameters CTGAN** The following hyperparameters were tuned: number of hidden layers of generator $\in \{1, 2, 3, 4\}$, number of nodes per hidden layer of generator $\in \{8, 16, 32, 64, 128, 256, 512\}$, number of hidden layers of discriminator $\in \{1, 2, 3, 4\}$, number of nodes per hidden layer of discriminator $\in \{8, 16, 32, 64, 128, 256, 512\}$, number of epochs log-uniformly $\in [5, 300]$, number of iterations in the discriminator per iteration of generator $\in \{1, 5, 10\}$, learning rate (the same for generator and discriminator) log-uniformly $\in [1e{-}6, 1e{-}2]$, dropout in generator uniformly $\in [0, 1]$, dropout in discriminator uniformly $\in [0, 1]$, weight decay of generator log-uniformly $\in [1e{-}6, 1]$, and weight decay of discriminator log-uniformly $\in [1e{-}6, 1]$. Batch size was fixed at $\min(200, n)$, with $n$ the sample size.

**Hyperparameters TVAE** The following hyperparameters were tuned: embedding dimension $\in \{32, 64, 128, 256, 512\}$, number of hidden layers (the same for encoder and decoder) $\in \{1, 2, 3, 4\}$, number of nodes per hidden layer (the same for encoder and decoder) $\in \{32, 64, 128, 256, 512\}$, number of epochs log-uniformly $\in [200, 1000]$, reconstruction error loss factor $\in \{1, 2, 5, 10\}$, dropout in encoder uniformly $\in [0, 1]$, dropout in decoder uniformly $\in [0, 1]$, and weight decay (the

same for encoder and decoder) log-uniformly $\in [1\mathrm{e}{-}6, 1]$. Batch size was fixed at $\min(200, n)$, with $n$ the sample size.

**Objective score**    The average inverse of the Kullback–Leibler divergence (IKLD) between the original and the synthetic dataset was used as metric. 5-fold cross-validation was used as follows: each time four training folds of original data were used to train the generative model and one validation fold of original data was used to calculate the IKLD with the synthetic data (of equal size as the validation fold) generated by the model. To make the IKLD metric independent of data dimension (lower IKLDs are typically seen for smaller sample sizes even if the synthetic datasets would be sampled directly from the ground truth population), the 5-fold cross-validated IKLD was normalised by the 5-fold cross-validated IKLD of a generative model that simply generates bootstrap samples of the original data. Finally, this procedure was repeated and averaged over five seeds to make the performance independent of seed initialisation (used for split in train and validation sets during cross-validation and for generative model initialisation). The objective score is thus the normalised 5-fold cross-validated IKLD averaged over five seeds. Note that we opted to use this score as the IKLD is a widely used measure, though this choice remains rather arbitrary. Although alternative tuning objectives could impact the convergence rate of the SEs, we expect that they remain still slower than 1 over root-$n$ due to the highly data-adaptive nature of deep generative models.

**Optimisation algorithm**    The `Optuna` package (Akiba et al., 2019) was used to optimise the objective score in the hyperparameter space. First 100 hyperparameter configurations were randomly sampled from the specified hyperparameter search space. Subsequently, the Tree-structured Parzen Estimator algorithm (a Bayesian optimisation algorithm that uses a Gaussian mixture model as surrogate model) was applied to propose promising hyperparameter configurations until the hyperparameter optimisation study exceeded 12 hours (on a single NVIDIA Ampere A100 GPU) for each generative model. To reduce computation costs, median pruning was enabled after 10 hyperparameter proposals: if the hyperparameter configuration proposed yielded a moving average of the normalised 5-fold cross-validated IKLD after $x$ seed initialisation(s) that was worse than the median of the moving average obtained by previous hyperparameter configurations after the same number of seed initialisations, then this hyperparameter configuration was discarded.

**Performance**    The optimisation study was performed for a random sample of size $n = 500$ from the population. 419 and 1812 hyperparameter configurations were proposed for CTGAN and TVAE, respectively, of which 280 and 1409 were discarded by the pruning algorithm. The top three configurations were then evaluated on a random sample of sizes $n = 50$ and $n = 5000$ to check the applicability of the proposed hyperparameter configurations to other sample sizes. Based on this, the following configuration was chosen for **CTGAN**: 3 hidden layers in the generator, each with 512 nodes, 3 hidden layers in the discriminator, each with 128 nodes, 58 epochs, 10 iterations in the discriminator per iteration of generator, learning rate of $1.7\mathrm{e}{-}5$ (the same for generator and discriminator), dropout in generator of $88.9\%$, dropout in discriminator of $38.4\%$, weight decay of generator of $6.9\mathrm{e}{-}6$, and weight decay of discriminator of $1.4\mathrm{e}{-}3$. The following configuration was chosen for **TVAE**: embedding dimension of 64, 1 hidden layer with 512 nodes each (the same for encoder and decoder), 961 epochs, loss factor of 10, dropout in encoder of $7.3\%$, dropout in decoder of $77.5\%$, and weight decay of $1.3\mathrm{e}{-}4$ (the same for encoder and decoder).

# C    SIMULATION STUDY

## C.1    DATA GENERATING MECHANISM

Inspired by an applied medical setting, we create a hypothetical disease, defined by a low-dimensional tabular data generation mechanism. The dependency structure depicted by the Directed Acyclic Graph in Figure 2 in the main text displays the presence of five variables, each of them chosen to obtain a mix of data types. In our hypothetical disease, it is assumed that a patient is observed at a given point in time. At this time, patient data about *age*, *disease stage*, *biomarker*, and the random assignment of *therapy* is gathered. The binary outcome variable *death* is evaluated at a later time point, making this design a simplification, since we do not consider the data as longitudinal.

The exact routine to reconstruct this data generating mechanism is presented in the pseudo-code in Algorithm 1. *Age* (continuous) follows a normal distribution with mean 50 and standard deviation 10. *Disease stage* (ordinal) was generated according to a proportional odds cumulative logit model where an increase in *age* causes an increase in the odds of having a *disease stage* higher than a given stage $k$ ($\nu_{age} = -0.05$ and intercepts $\nu_{stage} = \{2, 3, 4\}$ for stage I-III). The variable *biomarker* (continuous) is a quantification of the *disease stage* and was also based on a generalised linear model, where *biomarker* follows a gamma distribution and its mean changes in function of *disease stage*. It was constructed in such a way that a higher *disease stage* results in higher values for the *biomarker* ($\gamma_0 = 4$, $\gamma_{stage} = \{0, -1, -2, -3\}$ for stage I-IV, respectively). *Therapy* (binary) is considered to be 1:1 randomly assigned and is therefore sampled from a Bernouilli distribution with a probability of 0.50. The last variable, *death* (binary), is generated by using a binomial logistic regression model in which the odds of *death* increase with an increasing *age* ($\beta_{age} = 0.05$), a higher *disease stage* ($\beta_{stage} = \{0, 0.50, 1.00, 1.50\}$ for stage I-IV, respectively), and absence of *therapy* ($\beta_{therapy} = -0.50$).

---

**Algorithm 1:** Data Generating Mechanism for Hypothetical Disease.

---

**input**    : Requested number of data records $n$.
**output** : Dataframe $D$ with $n$ records, each made up of 5 attributes: $age$, $stage$, $biomarker$, $therapy$, $death$.

$D \leftarrow Empty\ dataframe$
**for** $i \leftarrow 1$ **to** $n$ **do**

   $age \leftarrow Normal(mean = 50, std = 10)$

   $\nu_{age} \leftarrow 0.05$
   $\nu_I, \nu_{II}, \nu_{III} \leftarrow 2, 3, 4$
   $cp_I, cp_{II}, cp_{III} \leftarrow Sigmoid(\nu_I - \nu_{age} \times age), Sigmoid(\nu_{II} - \nu_{age} \times age), Sigmoid(\nu_{III} - \nu_{age} \times age)$
   $stage \leftarrow Categorical(cat = [I, II, III, IV], probs = [cp_I, cp_{II} - cp_I, cp_{III} - cp_{II}, 1 - cp_{III}])$

   $\gamma_0 \leftarrow 4$
   $\gamma_I, \gamma_{II}, \gamma_{III}, \gamma_{IV} \leftarrow 0, -1, -2, -3$
   $biomarker \leftarrow Gamma(shape = 25, scale = \frac{1}{25 \times (\gamma_0 + \gamma_{stage})})$

   $therapy \leftarrow Categorical(cat = [False, True], p = [0.5, 0.5])$

   $\beta_{age}, \beta_{therapy} \leftarrow 0.05, -0.50$
   $\beta_I, \beta_{II}, \beta_{III}, \beta_{IV} \leftarrow 0, 0.50, 1.00, 1.50$
   $p_{death} \leftarrow Sigmoid(-3 + \beta_{age} \times age + \beta_{stage} + \beta_{therapy} \times therapy)$
   $death \leftarrow Categorical(cat = [False, True], p = [1 - p_{death}, p_{death}])$

   $D_i \leftarrow \{age, stage, biomarker, therapy, death\}$
**end**

---

## C.2    QUALITY OF SYNTHETIC DATA

We performed some additional analyses to assess the quality of the synthetic data obtained in our simulation study.

**Average IKLD**    The inverse of the Kullback-Leibler divergence (IKLD) between original and synthetic data, averaged over 200 Monte Carlo runs and standardised between 0 and 1, is presented in Table C1, where the tuned **CTGAN** and **TVAE** have higher IKLD than their default versions. However, the statistical approaches still seem to perform slightly better.

Table C1: The IKLD between original and synthetic data, averaged over 200 Monte Carlo runs and standardised between 0 and 1. Higher values indicate similar datasets in terms of underlying distribution.

| Generator | $n = 50$ | $n = 160$ | $n = 500$ | $n = 1600$ | $n = 5000$ |
|---|---|---|---|---|---|
| **Synthpop** | 0.939 | 0.976 | 0.994 | 0.995 | 0.996 |
| **BN** | 0.934 | 0.984 | 0.997 | 0.998 | 0.999 |
| **BN DAG** | 0.936 | 0.986 | 0.996 | 0.998 | 0.999 |
| **Default CTGAN** | 0.853 | 0.905 | 0.861 | 0.903 | 0.933 |
| **CTGAN** | 0.918 | 0.952 | 0.975 | 0.984 | 0.988 |
| **Default TVAE** | 0.822 | 0.861 | 0.915 | 0.959 | 0.983 |
| **TVAE** | 0.838 | 0.898 | 0.979 | 0.996 | 0.998 |

**Failed generators**   Our **Default CTGAN** model, as implemented in Synthcity, could not be trained in one run (run 51 for $n = 500$) due to an internal error in the package. As such, it was not possible to generate synthetic data with **Default CTGAN** in this run. This comprises $0.1\%$ ($1/1000$) of all **Default CTGANs** trained and $0.01\%$ ($1/8000$) of all generators trained. The other generative models did not produce errors during training, so that synthetic data could be generated in every run.

**Exact memorisation**   A sanity check was conducted to ensure that no records of the original data were memorised by the generative model. **BN** made the following number of exact copies of the original data in the synthetic data: one record ($2.00\%$) for $n = 50$ in five runs (runs 66, 103, 139, 147, 178), one record ($0.63\%$) for $n = 160$ in one run (run 55), and one record ($0.02\%$) for $n = 5000$ in one run (run 90). **BN DAG** made the following exact copies: one record ($2.00\%$) for $n = 50$ in two runs (runs 66, 103). The other generative models did not make exact copies.

**Non-estimable estimators**   Due to sparse data, especially for small sample sizes, some of the 17 estimators considered could not be estimated in a small subset of the 7999 (original and synthetic) datasets, producing extremely small ($< 1e{-}10$) or large ($> 1e2$) standard errors. Overall, $0.35\%$ ($481/135\,983$) estimates could not be obtained, mainly for $n = 50$ ($1.71\%; 466/27\,200$) and to a much lesser extent for $n = 160$ ($0.02\%; 6/27\,200$) and $n = 500$ ($0.03\%; 9/27\,183$). The number of estimates that could not be obtained are presented per estimator in Table C2 and per generator in Table C3 for each sample size.

Table C2: The number of estimates that could not be obtained (due to sparse data) in the (original or synthetic) sample per sample size $n$ and per estimator.

| | $n = 50$ | $n = 160$ | $n = 500$ | $n = 1600$ | $n = 5000$ | **All** |
|---|---|---|---|---|---|---|
| *Proportion* | | | | | | |
| Proportion stage II | 11 | 0 | 0 | 0 | 0 | **11** |
| Proportion stage III | 7 | 0 | 0 | 0 | 0 | **7** |
| Proportion stage IV | 16 | 0 | 0 | 0 | 0 | **16** |
| *Gamma regression* | | | | | | |
| Effect stage II on death | 11 | 0 | 0 | 0 | 0 | **11** |
| Effect stage III on death | 7 | 0 | 0 | 0 | 0 | **7** |
| Effect stage IV on death | 16 | 0 | 0 | 0 | 0 | **16** |
| *Logistic regression* | | | | | | |
| Effect therapy on death | 11 | 0 | 0 | 0 | 0 | **11** |
| Effect stage II on death | 91 | 1 | 5 | 0 | 0 | **97** |
| Effect stage III on death | 116 | 3 | 1 | 0 | 0 | **120** |
| Effect stage IV on death | 180 | 2 | 3 | 0 | 0 | **185** |
| All | 466 | 6 | 9 | 0 | 0 | **481** |

Table C3: The number of estimates that could not be obtained (due to sparse data) in the (original or synthetic) sample per sample size $n$ and per generator.

| | Original | Synthpop | BN | BN DAG | Default CTGAN | CTGAN | Default TVAE | TVAE | **All** |
|---|---|---|---|---|---|---|---|---|---|
| $n = 50$ | 19 | 48 | 57 | 4 | 137 | 23 | 91 | 87 | **466** |
| $n = 160$ | 0 | 0 | 0 | 0 | 5 | 0 | 1 | 0 | **6** |
| $n = 500$ | 0 | 0 | 0 | 0 | 9 | 0 | 0 | 0 | **9** |
| $n = 1600$ | 0 | 0 | 0 | 0 | 0 | 0 | 0 | 0 | **0** |
| $n = 5000$ | 0 | 0 | 0 | 0 | 0 | 0 | 0 | 0 | **0** |
| All | 19 | 48 | 57 | 4 | 151 | 23 | 92 | 87 | **481** |

## C.3 ADDITIONAL RESULTS

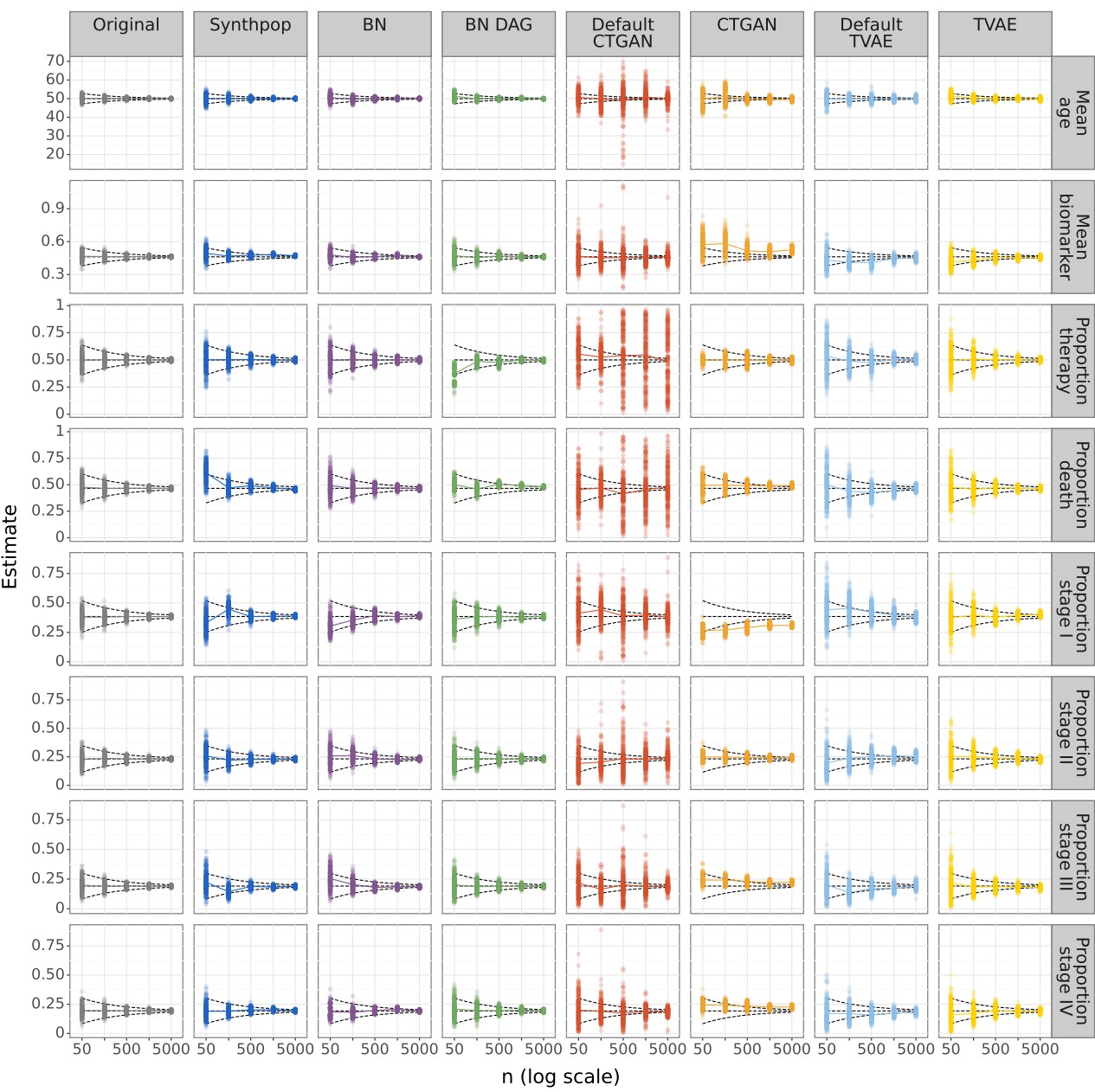

Figure C1: Simulation study results for all mean and proportion estimators. Each dot is an estimate per Monte Carlo run (200 dots in total per value of $n$). The population parameter is represented by the horizontal dashed line. The dashed funnel indicates the behaviour of an unbiased and $\sqrt{n}$-consistent estimator based on observed data.

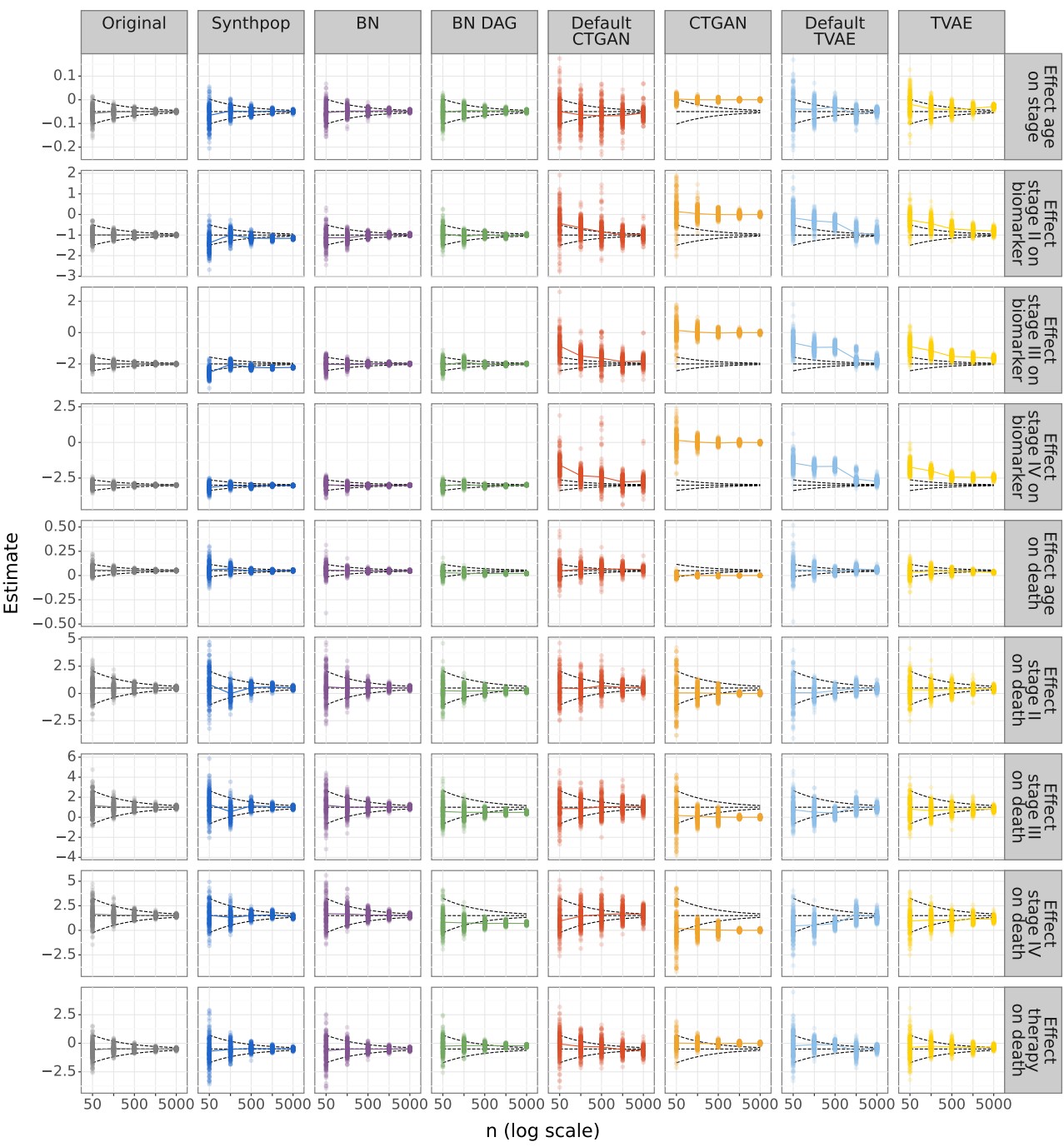

Figure C2: Simulation study results for all regression coefficient estimators. Each dot is an estimate per Monte Carlo run (200 dots in total per value of $n$). The population parameter is represented by the horizontal dashed line. The dashed funnel indicates the behaviour of an unbiased and $\sqrt{n}$-consistent estimator based on observed data.

Table C4: Relative error (RE) for all statistical approaches and all estimators, averaged over 200 Monte Carlo runs. $\text{RE}_{\hat{\theta}}$ is the relative bias of the estimates $\hat{\theta}$. $\text{RE}_{\hat{\sigma}_{\hat{\theta}}}$ is the relative error between the naive model-based ($\hat{\sigma}_{\hat{\theta},naive}$) and the empirical standard error. Positive and negative values indicate a relative over- and underestimation.

| | Synthpop | | | | BN | | | | BN DAG | | | |
| | $\text{RE}_{\hat{\theta}}$ (%) | | $\text{RE}_{\hat{\sigma}_{\hat{\theta}}}$ (%) | | $\text{RE}_{\hat{\theta}}$ (%) | | $\text{RE}_{\hat{\sigma}_{\hat{\theta}}}$ (%) | | $\text{RE}_{\hat{\theta}}$ (%) | | $\text{RE}_{\hat{\sigma}_{\hat{\theta}}}$ (%) | |
| **Estimator** | $n=50$ | $n=5000$ | $n=50$ | $n=5000$ | $n=50$ | $n=5000$ | $n=50$ | $n=5000$ | $n=50$ | $n=5000$ | $n=50$ | $n=5000$ |
|---|---|---|---|---|---|---|---|---|---|---|---|---|
| *Mean* | | | | | | | | | | | | |
| Mean age | 0.15 | 0.03 | -40.31 | -26.55 | 1.05 | 0.02 | -27.71 | -31.21 | 2.59 | -0.10 | -6.17 | -2.32 |
| Mean biomarker | 6.86 | 2.16 | -14.23 | 1.16 | 3.58 | -0.11 | -5.65 | 5.32 | 2.32 | 0.05 | -21.31 | -26.83 |
| *Proportion* | | | | | | | | | | | | |
| Proportion therapy | 0.18 | -0.12 | -33.26 | -31.47 | -0.92 | 0.09 | -33.05 | -33.94 | -25.28 | -0.93 | 7.00 | -4.56 |
| Proportion death | 31.39 | -2.35 | -25.40 | -14.07 | 5.76 | -0.26 | -29.85 | -34.33 | 7.27 | 3.23 | 45.38 | 25.23 |
| Proportion stage I | -14.22 | 0.83 | -27.41 | -7.91 | -21.00 | 1.51 | -9.42 | -18.97 | -5.89 | -0.02 | -24.20 | -29.00 |
| Proportion stage II | 11.15 | -0.77 | -31.54 | -23.17 | 11.68 | -0.13 | -28.55 | -26.40 | -2.22 | -0.07 | -24.85 | -33.21 |
| Proportion stage III | 17.03 | -1.48 | -30.17 | -25.81 | 32.74 | -2.86 | -27.63 | -19.00 | 5.57 | -0.26 | -30.35 | -22.50 |
| Proportion stage IV | -1.89 | 0.74 | -11.41 | -11.20 | -4.59 | -0.01 | 2.20 | -6.50 | 8.90 | 0.40 | -22.85 | -26.96 |
| *Cumulative regression* | | | | | | | | | | | | |
| Effect age on stage | 31.92 | -1.51 | -30.07 | -28.01 | 9.37 | -3.77 | -25.69 | -25.85 | 7.60 | -3.05 | -19.11 | -31.98 |
| *Gamma regression* | | | | | | | | | | | | |
| Effect stage II on biomarker | 38.07 | 15.61 | -14.43 | 3.93 | 12.84 | -1.63 | -26.98 | -15.29 | -5.45 | -1.75 | -11.56 | 2.40 |
| Effect stage III on biomarker | 27.82 | 11.13 | 9.38 | 14.45 | 1.08 | -0.19 | -26.03 | -11.83 | 3.92 | -0.27 | -16.07 | -3.35 |
| Effect stage IV on biomarker | 4.96 | 0.89 | 12.95 | 31.81 | -1.25 | -0.01 | -23.09 | -13.33 | 1.59 | -0.72 | 0.74 | -2.41 |
| *Logistic regression* | | | | | | | | | | | | |
| Effect age on death | 19.07 | 2.35 | -33.90 | -30.04 | 13.74 | -3.18 | -30.01 | -26.43 | -41.76 | -57.81 | -9.98 | -3.13 |
| Effect stage II on death | 62.75 | -1.30 | -29.50 | -24.99 | 20.75 | 2.10 | -21.82 | -29.32 | -49.14 | -54.56 | -12.22 | -6.64 |
| Effect stage III on death | 28.07 | -0.46 | -15.91 | -21.00 | 19.98 | 3.19 | -19.33 | -31.96 | -42.65 | -51.54 | -6.69 | 0.39 |
| Effect stage IV on death | 2.23 | -8.20 | 0.08 | -20.22 | 12.70 | 0.65 | -12.17 | -31.24 | -45.20 | -53.25 | -9.68 | -16.36 |
| Effect therapy on death | 38.53 | -2.74 | -32.78 | -30.45 | 31.88 | 0.71 | -27.53 | -31.80 | -50.16 | -55.24 | -11.63 | -15.83 |

Table C5: Relative error (RE) for all deep learning approaches and all estimators, averaged over 200 Monte Carlo runs. $\mathrm{RE}_{\hat{\theta}}$ is the relative bias of the estimates $\hat{\theta}$. $\mathrm{RE}_{\hat{\sigma}_{\hat{\theta}}}$ is the relative error between the naive model-based ($\hat{\sigma}_{\hat{\theta},naive}$) and the empirical standard error. Positive and negative values indicate a relative over- and underestimation.

| | Default CTGAN | | | | CTGAN | | | |
| | $\mathrm{RE}_{\hat{\theta}}$ (%) | | $\mathrm{RE}_{\hat{\sigma}_{\hat{\theta}}}$ (%) | | $\mathrm{RE}_{\hat{\theta}}$ (%) | | $\mathrm{RE}_{\hat{\sigma}_{\hat{\theta}}}$ (%) | |
| **Estimator** | $n=50$ | $n=5000$ | $n=50$ | $n=5000$ | $n=50$ | $n=5000$ | $n=50$ | $n=5000$ |
| *Mean* | | | | | | | | |
| Mean age | 1.51 | -0.44 | -66.29 | -95.54 | -0.03 | -0.73 | -46.38 | -78.85 |
| Mean biomarker | 1.11 | -0.12 | -60.86 | -92.56 | 24.14 | 13.87 | -35.77 | -76.00 |
| *Proportion* | | | | | | | | |
| Proportion therapy | 10.64 | -3.10 | -52.41 | -97.84 | -0.42 | -0.18 | 131.41 | -56.79 |
| Proportion death | -2.39 | 2.09 | -56.80 | -96.64 | 6.04 | 5.29 | -9.52 | -47.44 |
| Proportion stage I | 7.50 | -0.58 | -53.63 | -91.41 | -30.49 | -19.44 | 37.80 | -43.39 |
| Proportion stage II | -17.69 | 4.88 | -48.00 | -92.24 | 6.63 | 4.60 | 46.34 | -52.61 |
| Proportion stage III | 13.00 | -0.97 | -48.69 | -84.79 | 26.72 | 16.35 | 33.84 | -51.59 |
| Proportion stage IV | 2.79 | -3.74 | -54.53 | -81.30 | 26.37 | 17.06 | 48.08 | -47.06 |
| *Cumulative regression* | | | | | | | | |
| Effect age on stage | -0.13 | 11.58 | -40.15 | -85.57 | -105.97 | -100.03 | 13.91 | -3.23 |
| *Gamma regression* | | | | | | | | |
| Effect stage II on biomarker | -56.70 | -2.90 | -23.39 | -77.69 | -114.42 | -99.61 | -20.44 | -7.70 |
| Effect stage III on biomarker | -57.18 | -9.09 | -28.55 | -87.97 | -107.06 | -99.61 | -14.75 | -8.60 |
| Effect stage IV on biomarker | -47.37 | -9.21 | -40.31 | -91.84 | -104.96 | -99.62 | -22.89 | -7.25 |
| *Logistic regression* | | | | | | | | |
| Effect age on death | -8.24 | 21.74 | -38.49 | -78.71 | -96.42 | -97.81 | -9.41 | -1.18 |
| Effect stage II on death | 6.99 | 14.60 | -17.44 | -71.45 | -88.74 | -99.15 | -44.22 | -11.08 |
| Effect stage III on death | -14.20 | 7.92 | -20.60 | -74.80 | -82.06 | -99.44 | -43.57 | -18.69 |
| Effect stage IV on death | -36.57 | 10.05 | -16.64 | -78.24 | -87.18 | -99.48 | -48.96 | -9.38 |
| Effect therapy on death | -87.68 | 24.40 | -20.59 | -72.08 | -104.72 | -101.02 | 17.50 | -13.89 |

| | Default TVAE | | | | TVAE | | | |
| | $\mathrm{RE}_{\hat{\theta}}$ (%) | | $\mathrm{RE}_{\hat{\sigma}_{\hat{\theta}}}$ (%) | | $\mathrm{RE}_{\hat{\theta}}$ (%) | | $\mathrm{RE}_{\hat{\sigma}_{\hat{\theta}}}$ (%) | |
| **Estimator** | $n=50$ | $n=5000$ | $n=50$ | $n=5000$ | $n=50$ | $n=5000$ | $n=50$ | $n=5000$ |
| *Mean* | | | | | | | | |
| Mean age | 0.04 | 0.14 | -65.82 | -85.71 | 2.79 | -0.38 | -40.97 | -78.55 |
| Mean biomarker | -6.49 | -1.34 | -58.62 | -77.81 | -9.67 | -1.09 | -48.37 | -63.51 |
| *Proportion* | | | | | | | | |
| Proportion therapy | 6.40 | 1.26 | -57.92 | -65.27 | -1.16 | -0.14 | -46.33 | -42.72 |
| Proportion death | 7.03 | 0.15 | -58.67 | -66.89 | 2.72 | -0.48 | -48.78 | -41.14 |
| Proportion stage I | 14.86 | -2.10 | -53.17 | -71.80 | -2.17 | 4.75 | -42.91 | -50.70 |
| Proportion stage II | -15.73 | 8.37 | -53.19 | -77.58 | 9.11 | -2.04 | -46.27 | -56.65 |
| Proportion stage III | 4.73 | -0.04 | -49.99 | -77.58 | 13.68 | -6.59 | -52.44 | -57.94 |
| Proportion stage IV | -11.56 | -5.83 | -48.23 | -66.16 | -16.18 | -0.50 | -45.01 | -59.32 |
| *Cumulative regression* | | | | | | | | |
| Effect age on stage | -22.49 | -3.86 | -26.09 | -72.83 | -62.29 | -43.48 | -15.99 | -53.27 |
| *Gamma regression* | | | | | | | | |
| Effect stage II on biomarker | -83.35 | -4.19 | -4.47 | -80.60 | -73.01 | -19.87 | 2.13 | -60.54 |
| Effect stage III on biomarker | -67.21 | -9.37 | -14.38 | -80.47 | -55.66 | -18.20 | -21.37 | -62.77 |
| Effect stage IV on biomarker | -52.00 | -9.01 | -22.71 | -83.32 | -42.03 | -18.43 | -22.65 | -66.43 |
| *Logistic regression* | | | | | | | | |
| Effect age on death | 17.87 | 15.48 | -34.93 | -74.68 | -36.16 | -35.79 | -11.26 | -44.03 |
| Effect stage II on death | -85.24 | -10.42 | -21.16 | -65.43 | -28.67 | -8.57 | -10.28 | -52.84 |
| Effect stage III on death | -24.49 | -4.02 | -11.77 | -67.33 | -19.32 | -20.20 | -9.94 | -53.38 |
| Effect stage IV on death | -69.64 | -10.59 | -13.54 | -74.70 | -37.17 | -23.28 | -11.38 | -64.08 |
| Effect therapy on death | -56.22 | -18.13 | -27.70 | -68.74 | -26.43 | -35.56 | -24.90 | -56.18 |

Table C6: Estimated exponent $a$ for the power law convergence rate $n^{-a}$ for the empirical standard error (SE).

| | Original | Synthpop | BN | BN DAG | Default CTGAN | CTGAN | Default TVAE | TVAE |
|---|---|---|---|---|---|---|---|---|
| **Estimator, SE** | | | | | | | | |
| *Mean* | | | | | | | | |
| Mean age | 0.49 [0.47; 0.52] | 0.53 [0.50; 0.55] | 0.49 [0.45; 0.52] | 0.49 [0.47; 0.52] | 0.03 [-0.34; 0.41] | 0.40 [0.13; 0.66] | 0.23 [0.11; 0.36] | 0.22 [0.03; 0.42] |
| Mean biomarker | 0.48 [0.44; 0.53] | 0.51 [0.44; 0.58] | 0.51 [0.48; 0.53] | 0.49 [0.47; 0.51] | 0.12 [-0.10; 0.35] | 0.34 [0.26; 0.42] | 0.30 [0.17; 0.43] | 0.36 [0.29; 0.42] |
| *Proportion* | | | | | | | | |
| Proportion therapy | 0.50 [0.43; 0.56] | 0.50 [0.46; 0.54] | 0.48 [0.46; 0.51] | 0.46 [0.38; 0.54] | -0.18 [-0.37; 0.01] | 0.19 [-0.14; 0.52] | 0.46 [0.36; 0.55] | 0.51 [0.42; 0.61] |
| Proportion death | 0.51 [0.49; 0.53] | 0.52 [0.49; 0.54] | 0.50 [0.44; 0.55] | 0.48 [0.40; 0.55] | -0.09 [-0.28; 0.10] | 0.41 [0.28; 0.53] | 0.45 [0.40; 0.50] | 0.53 [0.44; 0.62] |
| Proportion stage I | 0.48 [0.43; 0.52] | 0.54 [0.51; 0.57] | 0.46 [0.41; 0.52] | 0.48 [0.43; 0.53] | 0.13 [-0.02; 0.29] | 0.28 [0.11; 0.44] | 0.38 [0.34; 0.41] | 0.47 [0.41; 0.53] |
| Proportion stage II | 0.48 [0.46; 0.51] | 0.51 [0.47; 0.56] | 0.51 [0.49; 0.53] | 0.47 [0.45; 0.49] | 0.06 [-0.17; 0.29] | 0.26 [0.13; 0.39] | 0.31 [0.26; 0.37] | 0.44 [0.39; 0.49] |
| Proportion stage III | 0.51 [0.46; 0.56] | 0.51 [0.41; 0.61] | 0.56 [0.47; 0.64] | 0.51 [0.45; 0.56] | 0.18 [-0.16; 0.52] | 0.29 [0.17; 0.41] | 0.31 [0.20; 0.42] | 0.47 [0.37; 0.57] |
| Proportion stage IV | 0.48 [0.44; 0.52] | 0.47 [0.38; 0.57] | 0.49 [0.42; 0.55] | 0.50 [0.46; 0.54] | 0.29 [0.16; 0.43] | 0.29 [0.16; 0.41] | 0.39 [0.35; 0.43] | 0.41 [0.38; 0.44] |
| *Cumulative regression* | | | | | | | | |
| Effect age on stage | 0.53 [0.49; 0.56] | 0.54 [0.48; 0.60] | 0.51 [0.48; 0.55] | 0.48 [0.41; 0.55] | 0.20 [0.06; 0.33] | 0.40 [0.30; 0.50] | 0.34 [0.20; 0.48] | 0.44 [0.38; 0.51] |
| *Gamma regression* | | | | | | | | |
| Effect stage II on biomarker | 0.51 [0.47; 0.56] | 0.54 [0.50; 0.58] | 0.55 [0.51; 0.60] | 0.55 [0.52; 0.59] | 0.27 [0.08; 0.46] | 0.55 [0.48; 0.61] | 0.18 [-0.04; 0.41] | 0.26 [0.23; 0.30] |
| Effect stage III on biomarker | 0.50 [0.47; 0.54] | 0.52 [0.46; 0.58] | 0.56 [0.52; 0.61] | 0.55 [0.52; 0.58] | 0.16 [0.00; 0.31] | 0.52 [0.44; 0.61] | 0.19 [-0.01; 0.39] | 0.32 [0.24; 0.40] |
| Effect stage IV on biomarker | 0.50 [0.47; 0.53] | 0.54 [0.49; 0.58] | 0.57 [0.50; 0.64] | 0.52 [0.49; 0.55] | 0.10 [-0.19; 0.39] | 0.55 [0.43; 0.67] | 0.15 [-0.08; 0.37] | 0.28 [0.14; 0.43] |
| *Logistic regression* | | | | | | | | |
| Effect age on death | 0.56 [0.49; 0.63] | 0.59 [0.45; 0.72] | 0.56 [0.45; 0.67] | 0.57 [0.55; 0.59] | 0.28 [0.12; 0.44] | 0.47 [0.31; 0.63] | 0.37 [0.13; 0.61] | 0.51 [0.45; 0.58] |
| Effect stage II on death | 0.52 [0.47; 0.57] | 0.56 [0.49; 0.63] | 0.55 [0.52; 0.57] | 0.55 [0.50; 0.59] | 0.30 [0.23; 0.36] | 0.68 [0.41; 0.95] | 0.34 [0.09; 0.60] | 0.37 [0.28; 0.47] |
| Effect stage III on death | 0.52 [0.46; 0.59] | 0.55 [0.47; 0.64] | 0.52 [0.48; 0.57] | 0.55 [0.50; 0.60] | 0.24 [0.13; 0.35] | 0.65 [0.38; 0.93] | 0.30 [0.05; 0.54] | 0.36 [0.27; 0.46] |
| Effect stage IV on death | 0.53 [0.48; 0.57] | 0.51 [0.43; 0.59] | 0.50 [0.49; 0.51] | 0.52 [0.47; 0.56] | 0.20 [0.14; 0.27] | 0.71 [0.44; 0.97] | 0.23 [-0.09; 0.56] | 0.34 [0.28; 0.40] |
| Effect therapy on death | 0.53 [0.48; 0.58] | 0.56 [0.49; 0.63] | 0.55 [0.47; 0.62] | 0.53 [0.46; 0.59] | 0.25 [0.19; 0.31] | 0.51 [0.40; 0.61] | 0.29 [-0.07; 0.66] | 0.40 [0.30; 0.50] |

Table C7: Estimated exponent $a$ for the power law convergence rate $n^{-a}$ for the empirical bias.

| Estimator, bias | Original | Synthpop | BN | BN DAG | Default CTGAN | CTGAN | Default TVAE | TVAE |
|---|---|---|---|---|---|---|---|---|
| | | | | | **Generator** | | | |
| *Mean* | | | | | | | | |
| Mean age | 0.64 [0.40; 0.89] | 0.38 [0.18; 0.58] | 0.86 [0.39; 1.32] | 0.45 [-0.61; 1.50] | 0.08 [-0.61; 0.78] | -0.42 [-1.62; 0.77] | -0.09 [-0.99; 0.81] | 0.54 [-0.37; 1.45] |
| Mean biomarker | 0.47 [0.17; 0.77] | 0.10 [-0.43; 0.63] | 0.52 [-0.40; 1.44] | 0.81 [0.20; 1.41] | 0.42 [-0.34; 1.18] | 0.18 [-0.07; 0.43] | 0.37 [-0.14; 0.88] | 0.47 [-0.03; 0.96] |
| *Proportion* | | | | | | | | |
| Proportion therapy | 0.42 [0.09; 0.76] | 0.12 [-0.52; 0.75] | 0.63 [0.13; 1.14] | 0.81 [-0.12; 1.74] | 0.17 [-0.18; 0.52] | 0.10 [-0.34; 0.54] | 0.29 [-0.17; 0.76] | 0.63 [0.05; 1.22] |
| Proportion death | 1.24 [0.64; 1.85] | 0.45 [-0.52; 1.43] | 0.43 [-0.47; 1.34] | 0.02 [-0.63; 0.67] | -0.06 [-0.88; 0.75] | 0.04 [-0.01; 0.09] | 0.69 [-0.29; 1.67] | 0.58 [-0.29; 1.44] |
| Proportion stage I | 0.65 [-0.02; 1.32] | 0.79 [-0.52; 2.09] | 0.72 [0.01; 1.42] | 0.88 [-0.41; 2.18] | 0.58 [-0.02; 1.19] | 0.11 [0.06; 0.17] | 0.49 [0.20; 0.78] | -0.12 [-0.53; 0.30] |
| Proportion stage II | 0.92 [0.49; 1.36] | 0.57 [0.46; 0.67] | 0.91 [-0.24; 2.05] | 0.76 [0.34; 1.17] | 0.41 [-0.48; 1.30] | 0.10 [0.02; 0.19] | -0.00 [-0.52; 0.51] | 0.34 [-0.02; 0.70] |
| Proportion stage III | 0.46 [-0.51; 1.43] | 0.57 [-0.12; 1.27] | 0.47 [0.14; 0.81] | 0.70 [0.40; 0.99] | 0.66 [0.31; 1.02] | 0.13 [0.06; 0.19] | 0.90 [-0.87; 2.66] | 0.11 [-0.42; 0.63] |
| Proportion stage IV | 0.21 [-0.10; 0.51] | 0.04 [-0.69; 0.77] | 1.01 [-0.76; 2.78] | 0.47 [-0.23; 1.17] | -0.12 [-0.72; 0.48] | 0.11 [0.07; 0.15] | 0.14 [-0.15; 0.42] | 0.72 [0.29; 1.15] |
| *Cumulative regression* | | | | | | | | |
| Effect age on stage | 0.76 [0.27; 1.25] | 0.55 [-0.28; 1.39] | 0.12 [-0.17; 0.42] | 0.17 [-0.10; 0.44] | -0.80 [-2.48; 0.88] | 0.01 [-0.01; 0.03] | 0.29 [-0.33; 0.92] | 0.05 [-0.41; 0.51] |
| *Gamma regression* | | | | | | | | |
| Effect stage II on biomarker | 0.05 [-0.71; 0.81] | -0.09 [-1.29; 1.12] | 0.54 [-0.11; 1.19] | 0.20 [-0.25; 0.64] | 1.02 [-0.51; 2.56] | 0.03 [-0.00; 0.06] | 0.70 [0.23; 1.17] | 0.30 [0.20; 0.40] |
| Effect stage III on biomarker | 0.02 [-0.87; 0.90] | -0.03 [-1.02; 0.95] | 0.35 [-0.33; 1.04] | 0.58 [0.09; 1.08] | 0.43 [0.17; 0.69] | 0.01 [-0.00; 0.03] | 0.46 [0.16; 0.75] | 0.25 [0.14; 0.36] |
| Effect stage IV on biomarker | 0.37 [-0.29; 1.03] | 0.34 [-0.52; 1.19] | 0.90 [-0.36; 2.15] | 0.15 [-0.05; 0.34] | 0.38 [0.15; 0.61] | 0.01 [-0.01; 0.02] | 0.40 [0.13; 0.67] | 0.20 [0.05; 0.34] |
| *Logistic regression* | | | | | | | | |
| Effect age on death | 0.76 [-0.21; 1.74] | 0.52 [-0.01; 1.05] | 0.16 [-0.48; 0.79] | -0.07 [-0.14; 0.01] | -0.17 [-0.60; 0.25] | -0.00 [-0.02; 0.01] | -0.08 [-0.50; 0.34] | -0.05 [-0.55; 0.45] |
| Effect stage II on death | 0.49 [-0.27; 1.25] | 0.81 [-0.00; 1.63] | 0.34 [-0.34; 1.01] | -0.01 [-0.06; 0.05] | -0.08 [-0.69; 0.52] | -0.03 [-0.05; -0.00] | 0.58 [0.10; 1.07] | 0.30 [0.08; 0.52] |
| Effect stage III on death | 0.57 [-0.31; 1.45] | 0.87 [0.14; 1.60] | 0.14 [-1.65; 1.92] | -0.03 [-0.08; 0.01] | 0.14 [-0.37; 0.65] | -0.04 [-0.07; -0.01] | 0.47 [-0.21; 1.15] | 0.03 [-0.15; 0.20] |
| Effect stage IV on death | 0.62 [0.45; 0.79] | -0.16 [-1.37; 1.05] | 0.63 [0.32; 0.93] | -0.03 [-0.06; -0.00] | 0.14 [-0.52; 0.80] | -0.03 [-0.05; -0.01] | 0.51 [0.07; 0.95] | 0.10 [0.05; 0.15] |
| Effect therapy on death | 0.70 [0.53; 0.87] | 0.52 [0.11; 0.93] | 0.20 [-2.46; 2.86] | -0.02 [-0.07; 0.03] | 0.34 [-0.01; 0.68] | 0.00 [-0.03; 0.04] | 0.29 [-0.03; 0.61] | 0.02 [-0.24; 0.27] |

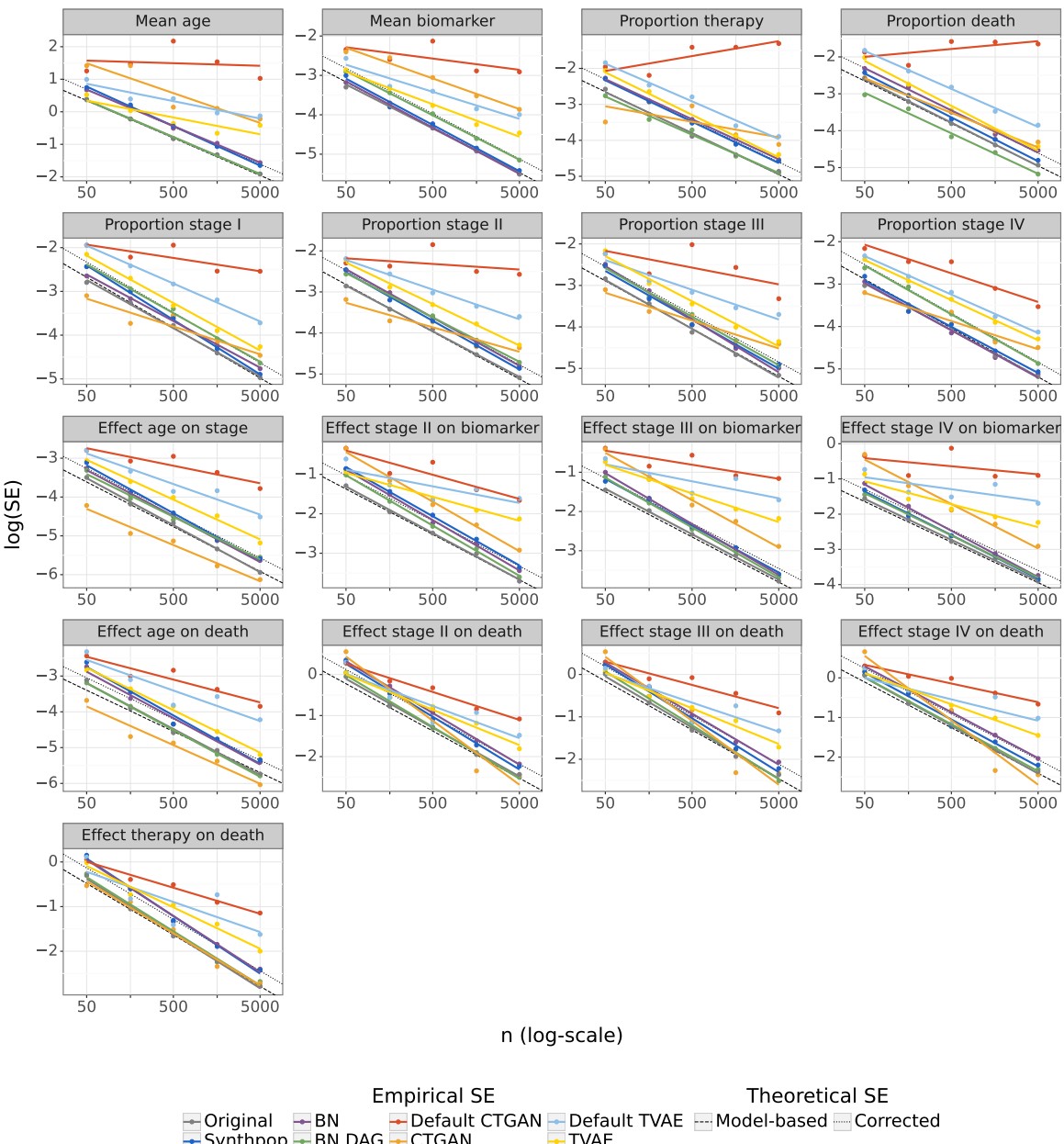

Figure C3: Convergence rate of the empirical standard error (SE). If the SE is of the form $\mathrm{SE} = cn^{-a}$, where $c$ is a constant, then $\log(SE) = \log c + (-a)\log(n)$. Therefore slope $a$ represents the convergence rate and the vertical offset $\log c$ the log asymptotic variance. The dashed line indicates the behaviour of the SE of an unbiased and $\sqrt{n}$-consistent estimator based on observed data, whereas the dotted line indicates the assumed behaviour of the SE of the same estimator based on synthetic data, following the correction proposed by Raab et al. (2016). Note that the asymptotic variances of the effect of *age* on *stage* and the effect of *age* on *death* by **CTGAN**, and the proportion of *death* by **BN DAG** are smaller than on the observed data, as they deliver a biased effect due to generative model misspecification.

# D  CASE STUDY

## D.1  QUALITY OF SYNTHETIC DATA

We performed some additional analyses to assess the quality of the synthetic data obtained in our case study.

**Average IKLD**  The inverse of the Kullback-Leibler divergence (IKLD) between original and synthetic data, averaged over 200 Monte Carlo runs and standardised between 0 and 1, is 0.923 for **Default CTGAN** and 0.955 for **Synthpop**.

**Failed generators**  Our **Default CTGAN** model, as implemented in Synthcity, could not be trained in three runs (runs $147, 156, 175$) due to an internal error in the package. As such, it was not possible to generate synthetic data with **Default CTGAN** in these runs. This comprises $1.50\%$ $(3/200)$ of all **Default CTGANs** trained and $0.75\%$ $(3/400)$ of all generators trained. **Synthpop** did not produce errors during training, so that synthetic data could be generated in every run.

**Exact memorisation**  A sanity check was conducted to ensure that no records of the original data were memorised by the generative model. **Synthpop** copied one original record $(0.02\%)$ in three runs (runs $5, 162, 197$). **Default CTGAN** did not make exact copies.

### D.1.1  Non-estimable estimators

The sample mean of $age$ and the sample effect of $age$ on $income$ (estimated via a logistic regression model) could be estimated in all (original and synthetic) datasets.

## D.2  ADDITIONAL RESULTS

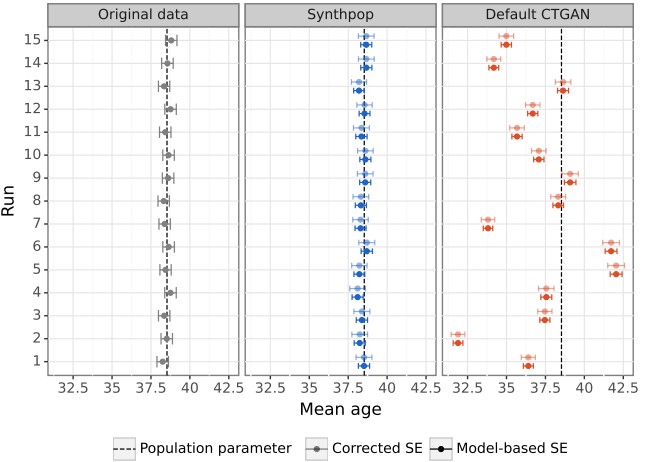

Figure D1: Empirical coverage of $95\%$ CIs for mean of $age$, with model-based and corrected standard error (SE).

# E    DIFFERENTIALLY PRIVATE GENERATORS

Here, we present the results for three state-of-the-art differentially private models: `PrivBayes` (J. Zhang et al., 2017), `DP-GAN` (Xie et al., 2018), and `PATE-GAN` (Jordon et al., 2018).

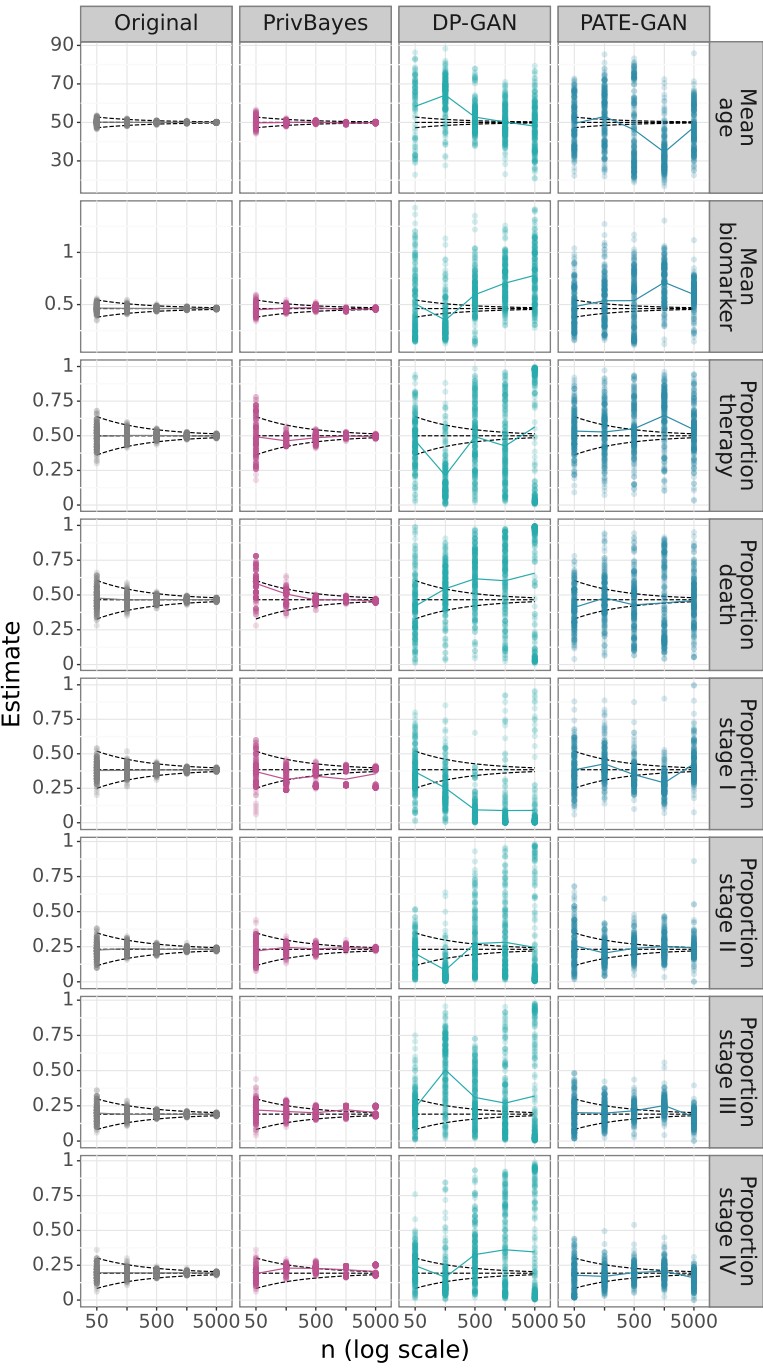

Figure E1: Simulation study results for all mean and proportion estimators. Each dot is an estimate per Monte Carlo run (200 dots in total per value of $n$). The population parameter is represented by the horizontal dashed line. The dashed funnel indicates the behaviour of an unbiased and $\sqrt{n}$-consistent estimator based on observed data.

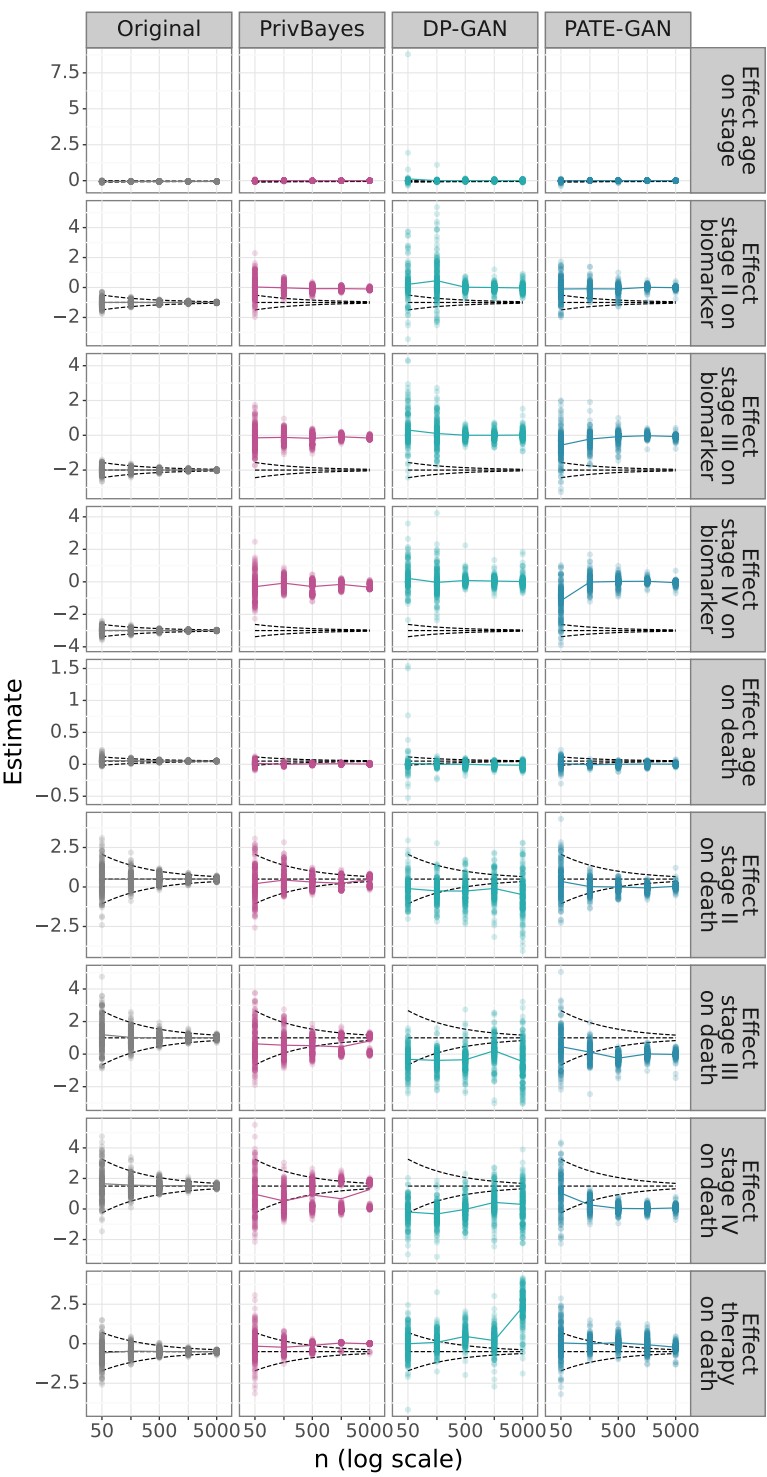

Figure E2: Simulation study results for all regression coefficient estimators. Each dot is an estimate per Monte Carlo run (200 dots in total per value of $n$). The population parameter is represented by the horizontal dashed line. The dashed funnel indicates the behaviour of an unbiased and $\sqrt{n}$-consistent estimator based on observed data.

Table E1: Estimated exponent $a$ for the power law convergence rate $n^{-a}$ for the empirical standard error (SE).

| | | Generator | | |
|---|---|---|---|---|
| **Estimator, SE** | **Original** | **PrivBayes** | **DP-GAN** | **PATE-GAN** |
| *Mean* | | | | |
| Mean age | 0.49 [0.47; 0.52] | 0.50 [0.27; 0.73] | 0.06 [-0.17; 0.30] | 0.04 [-0.18; 0.26] |
| Mean biomarker | 0.48 [0.44; 0.53] | 0.45 [0.37; 0.53] | 0.02 [-0.18; 0.22] | 0.07 [-0.11; 0.25] |
| *Proportion* | | | | |
| Proportion therapy | 0.50 [0.43; 0.56] | 0.60 [0.36; 0.83] | -0.18 [-0.30; -0.07] | -0.02 [-0.08; 0.03] |
| Proportion death | 0.51 [0.49; 0.53] | 0.58 [0.45; 0.70] | -0.14 [-0.31; 0.02] | -0.08 [-0.25; 0.09] |
| Proportion stage I | 0.48 [0.43; 0.52] | 0.09 [-0.08; 0.25] | -0.07 [-0.29; 0.15] | 0.05 [-0.11; 0.20] |
| Proportion stage II | 0.48 [0.46; 0.51] | 0.55 [0.41; 0.69] | -0.26 [-0.44; -0.09] | 0.10 [-0.07; 0.27] |
| Proportion stage III | 0.51 [0.46; 0.56] | 0.15 [0.01; 0.29] | -0.18 [-0.34; -0.02] | 0.14 [0.01; 0.27] |
| Proportion stage IV | 0.48 [0.44; 0.52] | 0.05 [-0.20; 0.30] | -0.26 [-0.29; -0.23] | 0.12 [-0.00; 0.25] |
| *Cumulative regression* | | | | |
| Effect age on stage | 0.53 [0.49; 0.56] | 0.52 [0.50; 0.54] | 0.60 [-0.12; 1.31] | 0.36 [0.01; 0.71] |
| *Gamma regression* | | | | |
| Effect stage II on biomarker | 0.51 [0.47; 0.56] | 0.53 [0.44; 0.61] | 0.41 [-0.04; 0.85] | 0.42 [0.29; 0.56] |
| Effect stage III on biomarker | 0.50 [0.47; 0.54] | 0.47 [0.35; 0.59] | 0.33 [-0.05; 0.70] | 0.45 [0.31; 0.59] |
| Effect stage IV on biomarker | 0.50 [0.47; 0.53] | 0.34 [0.21; 0.47] | 0.25 [-0.03; 0.53] | 0.46 [0.26; 0.66] |
| *Logistic regression* | | | | |
| Effect age on death | 0.56 [0.49; 0.63] | 0.34 [0.15; 0.54] | 0.27 [-0.19; 0.73] | 0.21 [-0.14; 0.55] |
| Effect stage II on death | 0.52 [0.47; 0.57] | 0.28 [0.12; 0.44] | -0.07 [-0.21; 0.07] | 0.38 [0.20; 0.55] |
| Effect stage III on death | 0.52 [0.46; 0.59] | 0.18 [0.03; 0.33] | -0.10 [-0.26; 0.07] | 0.36 [0.18; 0.54] |
| Effect stage IV on death | 0.53 [0.48; 0.57] | 0.08 [-0.09; 0.25] | -0.08 [-0.21; 0.04] | 0.42 [0.15; 0.69] |
| Effect therapy on death | 0.53 [0.48; 0.58] | 0.55 [0.39; 0.71] | -0.04 [-0.27; 0.19] | 0.25 [-0.03; 0.52] |

Table E2: Estimated exponent $a$ for the power law convergence rate $n^{-a}$ for the empirical bias.

| Estimator, bias | Original | Generator | | |
| --- | --- | --- | --- | --- |
| | | PrivBayes | DP-GAN | PATE-GAN |
| *Mean* | | | | |
| Mean age | 0.64 [0.40; 0.89] | 0.05 [-0.77; 0.87] | 0.63 [-0.45; 1.71] | -0.39 [-1.21; 0.43] |
| Mean biomarker | 0.47 [0.17; 0.77] | 0.46 [-0.30; 1.23] | -0.40 [-0.55; -0.26] | -0.45 [-0.89; -0.00] |
| *Proportion* | | | | |
| Proportion therapy | 0.42 [0.09; 0.76] | 0.37 [-0.58; 1.32] | 0.01 [-1.21; 1.23] | -0.19 [-0.67; 0.29] |
| Proportion death | 1.24 [0.64; 1.85] | 0.86 [-0.76; 2.49] | -0.29 [-0.47; -0.12] | 0.35 [-0.23; 0.92] |
| Proportion stage I | 0.65 [-0.02; 1.32] | -0.13 [-0.69; 0.43] | -0.58 [-1.20; 0.04] | -0.71 [-1.75; 0.33] |
| Proportion stage II | 0.92 [0.49; 1.36] | 0.15 [-0.41; 0.71] | 0.23 [-0.43; 0.90] | 0.18 [-0.25; 0.62] |
| Proportion stage III | 0.46 [-0.51; 1.43] | 0.11 [-0.31; 0.54] | -0.06 [-0.68; 0.56] | -0.30 [-0.85; 0.25] |
| Proportion stage IV | 0.21 [-0.10; 0.51] | -0.24 [-1.15; 0.67] | -0.33 [-0.75; 0.09] | -0.10 [-0.74; 0.53] |
| *Cumulative regression* | | | | |
| Effect age on stage | 0.76 [0.27; 1.25] | -0.03 [-0.12; 0.07] | 0.19 [-0.15; 0.53] | -0.08 [-0.27; 0.10] |
| *Gamma regression* | | | | |
| Effect stage II on biomarker | 0.05 [-0.71; 0.81] | 0.03 [0.01; 0.05] | 0.07 [-0.03; 0.17] | -0.03 [-0.05; 0.00] |
| Effect stage III on biomarker | 0.02 [-0.87; 0.90] | -0.00 [-0.02; 0.02] | 0.03 [-0.00; 0.06] | -0.06 [-0.12; 0.00] |
| Effect stage IV on biomarker | 0.37 [-0.29; 1.03] | 0.00 [-0.03; 0.04] | 0.01 [-0.01; 0.03] | -0.08 [-0.22; 0.05] |
| *Logistic regression* | | | | |
| Effect age on death | 0.76 [-0.21; 1.74] | -0.03 [-0.16; 0.09] | -0.12 [-0.19; -0.04] | -0.02 [-0.13; 0.09] |
| Effect stage II on death | 0.49 [-0.27; 1.25] | 0.36 [-0.50; 1.22] | -0.07 [-0.22; 0.08] | -0.21 [-0.56; 0.14] |
| Effect stage III on death | 0.57 [-0.31; 1.45] | 0.09 [-0.24; 0.43] | 0.03 [-0.18; 0.25] | -0.12 [-0.32; 0.08] |
| Effect stage IV on death | 0.62 [0.45; 0.79] | 0.16 [-0.26; 0.59] | 0.11 [0.00; 0.21] | -0.21 [-0.49; 0.07] |
| Effect therapy on death | 0.70 [0.53; 0.87] | -0.12 [-0.27; 0.02] | -0.32 [-0.65; 0.02] | 0.13 [-0.01; 0.27] |