# OpenReview forum: "The Real Deal Behind the Artificial Appeal: Inferential Utility of Tabular Synthetic Data"
_auai.org/UAI/2024/Conference — UAI 2024 spotlight_

### Official Review · Reviewer_GCrQ · 2024-03-19

**Q2-1 Originality-Novelty:** 2
**Q2-2 Correctness-Technical Quality:** 3
**Q2-5 Clarity Of Writing:** 3

**Q1 Summary And Contributions:**

The paper conducted an empirical study on the inferential utility of synthetic tabular datasets generated by statistical approaches and deep generative models. Specifically, the paper compared the results of the following statistical inference between real and synthetic datasets: the mean and standard error, the convergence rate of statistical estimator as sample size increases, the type I error rate in null hypothesis testing. The extensive experiments showed that the statistical inference on synthetic data are often more deviated from the real estimation, especially for deep generative models. The paper argues for closer scrutiny in statistical inference results based on synthetic data.

**Q2-3 Extent To Which Claims Are Supported By Evidence:**

3: Good: the main claims are supported by convincing evidence (in the form of adequate experimental evaluation, proofs, (pseudo-)code, references, assumptions).

**Q2-4 Reproducibility:**

3: Good: key resources (e.g. proofs, code, data) are available and key details (e.g. proofs, experimental setup) are sufficiently well-described for competent researchers to confidently reproduce the main results.

**Q3 Main Strengths:**

+ The paper is clearly written, provides strong motivation for its main subject and concisely explain all detains in the statistical inference performed.
+ The paper captures an interesting and important dimension of using synthetic tabular data. Despite many empirical evidence indicating the  ability of tabular synthesizers in preserving machine learning utility, the inferential utility has received less attention.
+ Solid adaption of prior work on inference with multiple synthetic datasets to one synthetic dataset case.
+ Details on reproducing results and well target visualization are included in the paper.

**Q4 Main Weakness:**

+ The paper need to make clearer distinction between statistical inference it discussed and the more general definition of fidelity in synthetic data. For example, one might attribute the high type I error in CTGAN synthetized data, as observed in section 4.2.4, to the failure of CTGAN in learning distribution of the training data. In other word, the paper should provide more evidence showing that such deviation in statistical inference is an inherent property of (deep) synthesizers, rather than just poor performance of one deep model.
+ The paper needs experiments on a wider range of deep generative models. DGMs is a fast-evolving domain and CTGAN is a model published almost 5 years ago. It would be better if the author include comparison on more recent state-of-the-arts generative models like diffusion-based ones and privacy-preserving generators that more closely aligned with the motivation of using synthetic data. While TVAE was tested in the paper, it was not consistently included for comparison when drawing conclusion on the flaws of deep generative models, which is confusing.

**Q5 Detailed Comments To The Authors:**

+ It would be great to see more discussion on the argument: "By means of a simulation study, we show that the rate of false positive findings (type 1 error) will be unacceptably high, even when the estimates are unbiased." But based on empirical evidences, the estimates made on synthetic data are biased as they does not capture the real means (especially for deep models).
+ Including more / better deep tabular synthesizer would make the results more convincing and up to date. Also include TVAE in the comparison for convergence rate and type I error.
+ Attributing problems in deep generative models to their regularization bias is an interesting idea and should have more discussion besides arguments.

**Q9 Complying With Reviewing Instructions:**

Yes

---

> ### Author Rebuttal · Authors · 2024-04-04
>
> We would like to thank **Reviewer GCrQ** for their careful reading of the manuscript and constructive review.
>
> *Make a clearer distinction between statistical inference and fidelity.*
>
> In line with the definition given in Jordon et al. (2022), we consider fidelity as measures that directly compare the synthetic dataset with the real dataset (rather than indirectly through a model or performance on a given task). With inferential utility of synthetic data, we are considering the question whether an inferential task can be solved in a valid way and would therefore classify this as a specific facet of utility rather than fidelity. We will include the distinction with fidelity in our revised manuscript.
>
> *Provide more evidence that deviation in statistical inference is an inherent property of (deep) synthesizers, rather than just poor performance of one deep model.*
>
> The CTGAN indeed partially fails to learn a valid distribution of the training data, translating to high bias in some of the estimators. As elaborated on in Section 4.2.2 in the main text, we observe both biased and unbiased estimators, though the decreased inferential utility of synthetic data generated by DGMs is noted regardless of whether the estimator is biased or not.
>
> We consider the minimal correction factor based on $\sqrt{n}$-consistent estimator an inherent property for statistical inference on synthetic data. However, we are currently designing deep neural network methods that are explicitly regularized during training to achieve an improved convergence compared to the models presented in the paper, and do not believe that the observed problems
> are inherent to well-designed deep models. Still, by evaluating four popular deep learning approaches, we argue that the naive training of current methods (optimizing maximum likelihood, with regularization for improved generalization), consistently seems to lead to poor convergence behavior.
>
> *The paper needs experiments on a wider range of deep generative models.*
>
> We agree that many more DGMs exist, and it would be interesting to investigate more recent diffusion-based models, or even LLMs. While these models are popular, they are less well-established in the domain of tabular synthetic data. We chose to focus on CTGAN and TVAE since these methods are often included in benchmarks in recent studies, and their implementation is more mature.
>
> We additionally want to thank the reviewer for bringing up differentially private (DP) generators, since this was not included in our original simulation study. Given the problematic behaviour of DGMs in terms of inferential utility, we did not originally include approaches that have a specific focus on privacy since these tackle an additional layer of complexity. In a revised version of our
> manuscript, we will include DP generators, but as expected, the same problems as observed with non-DP DGMs approaches persist. We refer to our rebuttal in reply to **Reviewer 7N2V** for the results.
>
> We would like to point out that due to the page restriction and to increase readability, we made a selection for the main text and refer the reader to Appendix C.3 and further for the results of all generators and estimators. All results for TVAE can be found in there, and are in line with the results described for CT-GAN and DGMs in general in the main text, but we will rewrite the results section to emphasize this.
>
> *The estimates made on synthetic data are biased as they do not capture the real means.*
>
> It is indeed true that some figures in the main text depict biased estimates. In Section C.3 in the appendix, we provide all estimates for all generators, and there we actually see many unbiased estimators, for both the statistical and deep learning generators. The main conclusion of our manuscript holds for all deep learning approaches, regardless of whether they delivered biased or unbiased estimators. Therefore, the low inferential utility cannot be explained in terms of bias or poor performance of one deep learning model. We thank the reviewer for this feedback and will draw attention to cases where the same shortcomings are observed even with unbiased estimates in our revised manuscript.
>
>
> *References*
>
> Jordon, J., Szpruch, L., Houssiau, F., Bottarelli, M., Cherubin, G., Maple, C.,... & Weller, A. (2022). Synthetic Data–what, why and how?.
> *arXiv preprint arXiv:2205.03257.*

---

### Official Review · Reviewer_1jXR · 2024-03-19

**Q2-1 Originality-Novelty:** 2
**Q2-2 Correctness-Technical Quality:** 3
**Q2-5 Clarity Of Writing:** 4

**Q1 Summary And Contributions:**

This paper considers doing population inference on synthetic data generated to protect the privacy of the original data sample. They use a simple, yet realistic, simulation setting and a real data example to show that naïve methods as well as a previously proposed standard error correction are not sufficient for valid inference on synthetic data.

**Q2-3 Extent To Which Claims Are Supported By Evidence:**

4: Excellent: all claims are supported by very convincing evidence (in the form of comprehensive experimental evaluation, rigorous mathematical proofs, detailed (pseudo-)code, precise references, well-motivated and realistic assumptions) and the authors deliver what they promise.

**Q2-4 Reproducibility:**

4: Excellent: key resources (e.g. proofs, code, data) are available and key details (e.g. proof sketches, experimental setup) are comprehensively described for competent researchers to confidently and easily reproduce the main results.

**Q3 Main Strengths:**

They identified and explored an important problem in a very illustrative manner. The paper was well motivated and well explained.

It was great to see the investigation happen on a simple yet realistic example and with a focus on doing inference (i.e. testing) as a downstream task. In this sense their analysis actually mimics what might be done in reality.

I could see their simulation setting becoming a benchmarking example for synthetic data generation and correction methods

**Q4 Main Weakness:**

The paper doesn't actually make an recommendations going forward. They show that standard methods don't work but make no attempt to even thing about what could be done to solve the problems.

It was not clear to me what their definition of 'privacy' was. I believe the standard is differential privacy (DP) but this was mentioned in the introduction and then not again. I am not sure the data generating methods they consider are DP, which raises the question how do they ensure 'privacy'? Is this simulation actually relevant if DP is what people are really concerned with?

**Q5 Detailed Comments To The Authors:**

A couple of references that I thought were related and not mentioned
+ ''Foundations of Bayesian Learning from Synthetic Data''
H Wilde, J Jewson, S Vollmer, C Holmes (2021) - this paper views learning from (Differentially Private) synthetic data as a model misspecification problems and establishes that naive inference would be biased
+ ''Mitigating Statistical Bias within Differentially Private Synthetic Data''
S Ghalebikesabi, H Wilde, J Jewson, A Doucet, SJ Vollmer, CC Holmes (2022) - this paper tries to debias inference from synthetic data using importance weights. It is not clear to me if this papear could be used in your setting but it feels worth mentioning either way.


Section 1

''Another approach that has recently been gaining attention is the use of synthetic data'' I was very surprised to see this presented as an alternative to DP. I had understood that synthetic data was usually generated to be DP. Have I misunderstood? Do the methods that are talked about in this paper - which aren't talked about as being DP - provide any privacy guarantees that they don't leak information about the underlying data they were trained on?

A few DP methods to generate synthetic data that would be good to mention when you mention DP and then also synthetic data
+ ''PrivBayes: Private data release via Bayesian networks'' Zhang, Cormode, Procopiuc, Srivastava, and Xiao.
+ ''Pate-gan: Generating synthetic data with differential privacy guarantees'' Jordon, Yoon, and van der Schaar.
+ ''Bayesian pseudocoresets'' Manousakas, Xu, Mascolo, and Campbell



Section 2.1

I was not particularly familiar with people talking about the ''regularisation bias'' for DL models. This is blamed for quite a lot of things. I wonder if some of the other poor performance of DL methods here could be attributed to either poor optimisation (not converging or finding a local mode) or actually just that they can't input the same useful structure that you can input into the statistical methods (i.e. the dependencies)

Section 2.2

What does ''$\sqrt{n}$-consistent (relative to the original data)'' mean?

Section 3.1 ''yet stochastic generative model $g(R)$'' what is stochastic about the generative model? Do you feed it some random inputs and it deterministically gives you outputs?

Section 4.1 ''We opted to work with low-dimensional tabular data given...'' if these methods don't work for a small example they are definitely not going to work for higher dimensional data. I did not really view this as  limitation of your paper.

''We intended a mix of ...'' this doesn't make sense. Maybe it should be ''We included...''

Section 4.2 Would releasing $m>n$ observations release more privacy? Some of the noise inherent in the synthetic data genertion could be removed by sampling more samples, ideally you have an infinite sample s (i.e. integrate over) and then weight the inference (somehow) to account for their only actually being $n$ for the standard errors.

I believe one has to be very careful with hyperparameter tuning when releasing synthetic data, as the tuning can break guarantees (e.g. DP ones) of the generator for example.

I was very confused by this inverse KLD distance, why is this somehow preferable to the standard KLD. It was then it was scaled, so the max (best) value was 1, but I have no idea if this corresponds to distributions being close or not. With KLD at least I know that KLD close to 0 means they must be very close. You say further ''This model mispecification is not captured well by the IKLD metric'' why are you using it then?

Just a thought, but a nice thing is that you can't correct your standard errors using something like bootstrap, which would be a standard procedure, because you can't resample the synthetic data.

Section 5 - I was a bit surprised that the affect of age on income was measured by a logistic regression, is income somehow binary here?

**Q9 Complying With Reviewing Instructions:**

Yes

---

> ### Author Rebuttal · Authors · 2024-04-04
>
> We would like to thank **Reviewer 1jXR** for their careful reading of the manuscript and constructive review.
>
> *The paper doesn't actually make any recommendations going forward.*
>
> For parametric approaches, the correction to the standard error (SE) previously proposed is sufficient and results in a nominally controlled type 1 error rate (only in absence of generative model misspecification). By contrast, such a practical and generic correction is hard to obtain for deep generative models (DGMs), since the uncertainty associated with their regularisation bias cannot readily be expressed analytically. Therefore, at present, this renders them not useful for statistical inference, despite their flexibility to better approximate the joint distribution of the original data. Rather than making specific recommendations, we aim to raise awareness for the potential problems associated with a statistical analysis of synthetic data, especially when generated by DGMs.
>
> To fully leverage the inferential utility of synthetic data created by DGMs, we propose the following ideas for future research:
>
> 1) Developing neural network based architectures and training strategies that result in new DGMs that offer better control of the uncertainty w.r.t. the estimand.
>
> 2) Developing targeted generation strategies that optimise the bias-variance trade-off of current DGMs w.r.t. the estimand by drawing insights from the literature on debiased and targeted machine learning.
>
> *1) It was not clear to me what their definition of 'privacy' was. 2) A few DP methods to generate synthetic data that would be good to mention.*
>
> Both privacy and utility are essential for synthetic data and their trade-off should be optimised. In our study, inferential utility was the starting point and as such, we did not formally define and assess privacy. By imposing differential privacy (DP) as an additional constraint during model training, this class of generative models provides a formal privacy protection. Initially, we did not include DGMs that focus specifically on privacy since these also need to tackle this additional layer of complexity and we demonstrate that non-DP DGMs already have low-to-no inferential utility. Following your and Reviewer 7N2V's suggestions, we included results for three state-of-the-art DP models (PrivBayes, PATE-GAN, and DP-GAN) in our reply to **Reviewer 7N2V**, noting that this did not impact the conclusions of our study. We admit that the distinction between synthetic data and DP was confusingly stated in the introduction, and we will rephrase this in the revised manuscript.
>
> *A couple of references that I thought were related and not mentioned.*
>
> We thank the reviewer for these highly relevant references and we will cover them in the revised manuscript.
>
> *What does "$\sqrt{n}$-consistent (relative to the original data)'' mean?*
>
> This means that the estimators converge to the population parameter at a $\sqrt{n}$ rate (with $n$ the sample size of the original data), in the sense that the difference between both is of the order 1 over $\sqrt{n}$.
>
> *What is stochastic about the generative model?*
>
> Indeed, each generative model is stochastic in the sense that there is some randomness in the sampling process, for both the statistical methods and the DGMs. In DGMs, additional randomness arises during the training process, due to weight initialisation and batching.
>
> *Would releasing $m>n$ observations release more privacy and reduce some of the noise inherent in the synthetic data?*
>
> Previous research has shown that the risk of disclosure increases with the number (or equivalently, sample size) of synthetic datasets. A synthetic sample size of $m\rightarrow+\infty$ would indeed deliver corrected SEs of estimators that are (asymptotically) equivalent to the original SEs, but only when these SEs keep the typical shrinkage with sample size of 1 over $\sqrt{n}$ rate. Our study indicates that DGMs fail to offer this specific guarantee.
>
> *I believe one has to be very careful with hyperparameter tuning.*
>
> Our main conclusions hold for both versions of our DGMs: one with default and one with tuned hyperparameters. However, we did not investigate the effect of hyperparameter tuning on (differential) privacy, and agree that this may pose a risk.
>
> *I was very confused by this inverse KLD distance.*
>
> This was an arbitrary choice made a priori as the KLD is a widely used measure. By default, the Synthcity package calculates the average inverse of the KLD and standardises it between $0$ and $1$, which indicates that the datasets are from different vs. similar distributions, respectively. Although alternative tuning objectives could impact the convergence rate of the SEs, we expect that they remain still slower than 1 over $\sqrt{n}$ due to the highly data-adaptive nature of current DGMs.
>
> *The effect of age on income was measured by a logistic regression, is income binary here?*
>
> Indeed, income is a binary variable in the Adult dataset.

---

### Official Review · Reviewer_oA1u · 2024-03-25

**Q2-1 Originality-Novelty:** 1
**Q2-2 Correctness-Technical Quality:** 2
**Q2-5 Clarity Of Writing:** 2

**Q1 Summary And Contributions:**

This paper investigates the inferential utility of the tabular synthetic data. In particularly, this paper aims to show theoretically, and mainly empirically, the native inference, even under unbiased estimation, yields high false-positive results.

**Q2-3 Extent To Which Claims Are Supported By Evidence:**

3: Good: the main claims are supported by convincing evidence (in the form of adequate experimental evaluation, proofs, (pseudo-)code, references, assumptions).

**Q2-4 Reproducibility:**

2: Fair: key resources (e.g. proofs, code, data) are unavailable but key details (e.g. proof sketches, experimental setup) are sufficiently well-described for an expert to confidently reproduce the main results.

**Q3 Main Strengths:**

-	This paper investigates an interesting and practical issues with great empirical potential.
-	This paper provides empirical evidence against the native inference with a carefully designed experimental framework.

**Q4 Main Weakness:**

-	The contributions of this paper seem to be incremental: the theoretical results Eq. (1) is already proposed in prior work. I would expect if there were any theoretical analysis or findings back up the empirical discovery (high false-positive rate).
-	The experiment settings employ existing data generation methods and evaluation metrics. The experiments are conducted on simple datasets.

**Q5 Detailed Comments To The Authors:**

- I am confused about the contribution of this paper in section 2.2. Please correct me if I am wrong. According to the authors, Raab et al., 2016 proposed the Eq. (1) but did not provide assumptions and derivation. This paper provides the required assumption ($\sqrt{n}$-consistency) and a detailed derivation to an existing conclusion (Eq. (1)). This feels like a trivial contribution.
- What’s the exact actions in Step 3 of general experimental framework in Figure 1? Can you elaborate on ‘learn a representation of original data distribution’?
- The authors discuss different methods to generate data using DAGs and BNs in section 3.2. I have a few questions:

    - What’s the algorithmic difference between the first implementation and second implementation? It feels like both implementation/methods estimate the condition mechanism in the given DAG/BN structures by minimizing likelihood.
    - Does the difference between second and third implementation lie in the availability of BN structure?
    - If the Chow-Liu algorithm cannot reveal the ground-truth DAG, then what’s the purpose of third implementation if it is known that the structure and parameters of BN are wrong?
    - I think the presentation in this section should be improved. A summary of the major technical components for each implementation should be provided.
- Experiments: The experiments are performed on two datasets: a medical dataset and Adult Census Income dataset. Those two datasets are relatively simple (small number of variables, simple underlying BN structure).
    - Can you observe the similar empirical results on complex datasets?
    - How will the complexity of datasets affect the empirical results?

**Q9 Complying With Reviewing Instructions:**

Yes

---

> ### Author Rebuttal · Authors · 2024-04-04
>
> We would like to thank **Reviewer oA1u** for their careful reading of the manuscript and constructive review.
>
> *Reproducibility: key resources (e.g. proofs, code, data) are unavailable.*
>
> Reproducibility was appreciated by the other reviewers, given the code in the supplementary material, and the extensive information in the appendices to reproduce the reported results. Which key resources are still lacking?
>
> *The contributions of this paper seem to be incremental: the theoretical results Eq. (1) is already proposed in prior work.*
>
> The minimal correction for the standard error has indeed been previously introduced in Raab et al., as we acknowledge (even though we provide a more general proof). However, the key message of our study is that even this correction is far too optimistic for synthetic data obtained via deep generative models (DGMs) as they imply much slower convergence rates of estimators, as we show empirically. Our study thus aims to generate awareness that standard statistical analyses of synthetic data can be very misleading (even when such correction factors are used) and that efforts to circumvent this are essential.
>
> *The experiment settings employ existing data generation methods and evaluation metrics.*
>
> We indeed do not propose new methods, but rather focus on what is (not yet) possible with existing methods. Our modus operandi originates from a statistical perspective, which focuses on typical characteristics of an estimator: bias, standard error, type 1 error and power. In this study, we evaluate the validity of estimators that are well-established on original data, but have remained understudied in synthetic data, especially when created by DGMs.
>
> *What's the exact actions in Step 3 of general framework in Figure 1?*
>
> This depends on the generating method that is used. The different generators are listed in Section 3.2, and a concise yet comprehensive elaboration on these different techniques is provided in Appendix B. Due to the page restriction, it was not possible to include this in the main text.
>
> *What’s the algorithmic difference between the first implementation and second implementation?*
>
> There are multiple differences between Synthpop and the Bayesian Network (BN) implementation. In a BN, a joint probability is obtained through factorization. When all variables are discrete, natural estimates for the Conditional Probability Distributions (CPDs) are the relative frequencies, which coincide with the MLE of a multinomial model. Within our BN implementation, continuous variables undergo discretisation. It is possible to avoid this, but practically this means imposing a linear Gaussian CPD for all variables, including the discrete ones, which undermines the representation power of the BN. In Synthpop, the joint distribution is also defined in terms of a series of conditional distributions. With its parametric methods, Synthpop imposes a specific distribution and parametric regression model depending on the variable type. Therefore, the likelihood can now be written as a function of these regression models, instead of just the multinomial likelihood function seen in BN, and the corresponding parameters of these regression models are estimated via MLE. Depending on the variable type, different parametric models are possible, as opposed to BNs, where the distribution is either multinomial for discrete variables or Gaussian when (non-discretized) continuous variables are included in the mix. Thus, the difference between Synthpop and BN lies in the flexibility of the assumed parametric distribution and the way each method deals with mixed variable types.
>
> *Does the difference between second and third implementation lie in the availability of BN structure?*
>
> Indeed, the difference lies in the fact that the latter receives a pre-specified Directed Acyclic Graph (DAG) fixing its dependency structure, while the former learns this DAG from the data. We included both BN and BN DAG in order to investigate whether the availability of the correct DAG would result in better performance (e.g. less variability in estimators) compared with a BN that does not have prior knowledge and needs to rely on data-adaptive DAG discovery. In many practical settings, the (full) DAG cannot be provided upfront since causal relationships between the variables are unknown. In those cases, dependency structure discovery methods like the Chow-Liu algorithm are often used to recover the DAG. For this reason, we wanted to include a model that follows this paradigm, even if its performance is upper bounded by a model that receives the DAG upfront.
>
> We thank the reviewer for these questions, explicitly adding these aspects to Appendix B.1 will improve readability.
>
> *The experiments are conducted on simple datasets.*
>
> If these methods don't even work on simple examples, we should not expect the issues addressed in this study to disappear upon scaling up the datasets, as was also noted by **Reviewer 1jXR**.

---

### Official Review · Reviewer_7N2V · 2024-03-25

**Q2-1 Originality-Novelty:** 3
**Q2-2 Correctness-Technical Quality:** 3
**Q2-5 Clarity Of Writing:** 4

**Q1 Summary And Contributions:**

The paper explores whether statistical estimators evaluated on synthetic, generated datasets reliably match estimators evaluated on the original dataset. Using synthetic and real data, the paper shows that they do not, and introduces a corrected approximation for the standard error of an estimator applied to synthetic data.

**Q2-3 Extent To Which Claims Are Supported By Evidence:**

4: Excellent: all claims are supported by very convincing evidence (in the form of comprehensive experimental evaluation, rigorous mathematical proofs, detailed (pseudo-)code, precise references, well-motivated and realistic assumptions) and the authors deliver what they promise.

**Q2-4 Reproducibility:**

4: Excellent: key resources (e.g. proofs, code, data) are available and key details (e.g. proof sketches, experimental setup) are comprehensively described for competent researchers to confidently and easily reproduce the main results.

**Q3 Main Strengths:**

1.	The work is clearly motivated and well written.
2.	The evaluation approach is clear and the results support the central claims of the paper.
3.	The paper derives a corrected approximation for the SE of an estimator

**Q4 Main Weakness:**

1.	What are the broader implications of this work? If a researcher is deciding whether to release synthetic versus differentially-private data to enable research on private data, what considerations should they prioritize?

**Q5 Detailed Comments To The Authors:**

1.	The paper uses the citation of Drechsler & Haensch, (2023) to support the claim that “inferential utility is often unmentioned, especially in the DL community.” While that specific paper is a survey of sorts, it seems like this is an important claim motivating your paper and deserves a bit more support. How much overlap is there between the kinds of utility mentioned in Drechsler & Haensch and inferential utility as you define it?
2.	Do you see any potential for constrained synthetic data generation that explicitly maintains inferential utility?


Section 2.2 typo: ``fixed fraction $\in ]0, 1]$’’

**Q9 Complying With Reviewing Instructions:**

Yes

---

> ### Author Rebuttal · Authors · 2024-04-04
>
> We would like to thank **Reviewer 7N2V** for their careful reading of the manuscript and constructive review. We are pleased to read about their appreciation concerning the novelty and contribution of our proposed work.
>
> *What are the broader implications of this work? If a researcher is deciding whether to release synthetic versus differentially-private data to enable research on private data, what considerations should they prioritize?*
>
> The main goal of our study is to raise awareness that naive statistical analyses on synthetic data have reduced inferential utility and that corrections previously proposed do not suffice for deep generative models. In particular, the first implication is that results from such naive analysis should be interpreted with caution. Ideally, a second implication would be that an adequate correction for the standard error of an estimator is used. However, we have shown that the current correction factors are not capable of capturing all added variability inherent to synthetic data generated by deep generative models. Therefore, an applied researcher could opt to use a parametric generation method, since the corrected standard errors are proven to be sufficient only in these settings.
>
> We want to thank the reviewer for bringing up differentially private (DP) generators, since this was not included in our simulation study. It was our goal to investigate different generators, ranging from parametric to (non-DP) deep generative models. Given the problematic behaviour of the latter in terms of inferential utility, we did not yet include deep generative models that focus specifically on privacy since these need to tackle an additional layer of complexity. The generators discussed in our manuscript do not provide any rigorous privacy guarantees whereas DP generative models do. In the revised manuscript, we will include DP generative models (PrivBayes, PATE-GAN, and DP-GAN), but we see the same problems as observed with (non-DP) deep generative models. A preview of these results is available from the following figure (https://assets.nicepagecdn.com/0cf50c0c/4896361/images/convergence_se_dp_methods.png) and table (https://assets.nicepagecdn.com/0cf50c0c/4896361/files/bias_dp_methods.pdf).
>
> *How much overlap is there between the kinds of utility mentioned in Drechsler \& Haensch and inferential utility as you define it?*
>
> In the paper of Drechsler & Haensch (2023), a highly qualitative summary is given concerning the evolution and recent developments of synthetic data. They divide the utility metrics in broadly three categories: global utility metrics, outcome-specific utility metrics and fit-for-purpose measures. Evaluating the inferential utility of a synthetic dataset is an outcome-specific utility metric and the confidence interval overlap measure proposed by Karr et al. (2006) can be used for this purpose. Their measure indicates to which degree the confidence interval based on the synthetic data overlaps with the corresponding one on the original data. Our study focuses on the validity of the synthetic confidence interval itself without directly comparing it to the original data. We show by means of a Monte Carlo simulation study that this validity is not guaranteed. So, the purpose was not to propose another utility measure, but in fact to evaluate the inferential utility itself for different generative models. We plan to make this more clear in the revised manuscript to prevent future confusion about this distinction.
>
> *Do you see any potential for constrained synthetic data generation that explicitly maintains inferential utility?*
>
> For parametric approaches, the correction to the standard error previously proposed is sufficient and results in a nominally controlled type 1 error rate (only in absence of generative model misspecification). However, the use of this corrected standard error will decrease the power, reflecting the loss of information when working with synthetic data. By contrast, such a practical and generic correction is hard to obtain for deep generative models, since the uncertainty associated with their regularisation bias cannot readily be expressed analytically. Instead, it seems sensible to tackle certain objectives during training, for example by adding a regularisation objective to the standard GAN loss that reduces the bias in a particular target estimator on a per-batch basis.
>
> *References*
>
> Drechsler, J., and Haensch, A.-C. (2023). 30 years of synthetic data. arXiv preprint, arXiv:2304.02.
>
> Karr, A. F., Kohnen, C. N., Oganian, A., Reiter, J. P., and Sanil, A. P. (2006). A framework for evaluating the utility of data altered to protect confidentiality. The American Statistician, 60(3):224-232.

---

### Official Review · Reviewer_EvtN · 2024-03-26

**Q2-1 Originality-Novelty:** 3
**Q2-2 Correctness-Technical Quality:** 3
**Q2-5 Clarity Of Writing:** 4

**Q1 Summary And Contributions:**

The paper targets the issue of synthetic tabular datasets. In particular, it highlights the importance of inferential utility and discovers the positive findings are too high.

**Q2-3 Extent To Which Claims Are Supported By Evidence:**

3: Good: the main claims are supported by convincing evidence (in the form of adequate experimental evaluation, proofs, (pseudo-)code, references, assumptions).

**Q2-4 Reproducibility:**

3: Good: key resources (e.g. proofs, code, data) are available and key details (e.g. proofs, experimental setup) are sufficiently well-described for competent researchers to confidently reproduce the main results.

**Q3 Main Strengths:**

The paper is well-written and focuses on an important problem. In particular, the paper focuses on the bias and standard error of the synthetic tabular datasets.

**Q4 Main Weakness:**

Although the paper has outlined well, the novelty is somewhat limited. The experimental section only shows a case study and did not compare with other methods.

**Q5 Detailed Comments To The Authors:**

Could you please elaborate more on why choose bias and standard error?

**Q9 Complying With Reviewing Instructions:**

Yes

---

> ### Author Rebuttal · Authors · 2024-04-04
>
> We would like to thank Reviewer **EvtN** for their careful reading of the manuscript and constructive review.
>
> *Could you please elaborate more on why choose bias and standard error?*
>
> When an inferential statement is made, we rely on the test statistic and its properties to obtain a p-value. The formula for a test statistic is typically the obtained *estimate* divided by its *standard error*. Therefore, since the objective of our study was evaluating the validity of inferential statements, we start with evaluating the estimate (and its corresponding bias) and the standard error of the estimator.
>
> *The experimental section only shows a case study and did not compare with other methods.*
>
> In our study, we first conducted an extensive simulation study (presented in Section 4) according to the experimental setup outlined in Section 3. In this simulation study, we compare seven data generating methods: Synthpop, a Bayesian Network with and without DAG pre-specification, a default CTGAN, a default TVAE, and tuned versions of both the CTGAN and TVAE. Due to the page limit and to enhance readability, we opted to only display the results for three generators in Figure 3 and Tables 1 and 2. However, we included the results for all data generating methods in the Appendix, Section C.3. In addition, we will include differentially private generators in the revised manuscript and show that also here the main conclusions of our paper hold, i.e. low inferential utility with deep generative models (see our reply to Reviewer **7N2V**).
>
> To illustrate our findings and its implications for an applied researcher, we provided a case study in Section 5 where we again follow the experimental setup outlined in Section 3, but now on a real-world dataset. For this case study, we restricted the analyses to two generative models, i.e. Synthpop and CTGAN, one model per category of generative models as suggested by Hernandez et al. (2022).
>
> *Although the paper has outlined well, the novelty is somewhat limited.*
>
> Our work stands out w.r.t. other contributions, such as the work of Raab et al. (2016), in that we are the first (1) to quantify the convergence rate of various estimators and generators, (2) to show the impact of deviant behavior on inferential utility, and (3) to empirically point out the detrimental bias-variance trade-off w.r.t. the estimand for deep generative models as compared to statistical approaches. With our work, we aim to raise awareness that the current correction factors for the standard error of an estimator are not routinely capable of capturing all added variability inherent to synthetic data, leading to a strong increase of false-positive findings (type 1 error), especially with deep generative models.
>
> *References*
>
> Hernandez, M., Epelde, G., Alberdi, A., Cilla, R., & Rankin, D. (2022). Synthetic data generation for tabular health records: A systematic review. Neurocomputing, 493, 28–45.
>
> Raab, G. M., Nowok, B., & Dibben, C. (2016). Practical data synthesis for large samples. Journal of Privacy and Confidentiality, 7 (3), 67–97.

---

### Meta-Review · Area_Chair_Ep2G · 2024-04-18

The paper is an empirical study of synthetic data in the context of tabular datasets. Reviewers commended the careful simulation set up and rigorous analysis, alongside the clear presentation. The paper further proposes a correction for standard error estimates derived in studies using synthetic data. I believe this paper is likely to have an impact on a wide variety of research being done these days with the aid of synthetic data.